# Context-aware Prompt Tuning: Enhancing Few-Shot Learning via Optimized Context Embeddings

## Abstract

Large Language Models (LLMs) can perform few-shot learning using either In-Context Learning (ICL) or optimization-based methods. While ICL typically excels in low-data regimes, optimization-based methods tend to perform better when more data is available. This contrast raises an important question: Why do optimization-based methods struggle in low-data scenarios, and how can they be effectively combined with ICL to enhance few-shot learning? In this work, we identify overfitting as the primary limitation of optimization-based methods in few-shot learning. To address this, we propose Context-Aware Prompt Tuning (CPT), which extends ICL through a carefully designed optimization process specifically crafted to mitigate overfitting. CPT extracts richer insights from limited data while preserving the integrity of the original input samples. We validate our approach across diverse classification and question answering tasks and multiple LLM architectures. CPT consistently outperforms existing baselines across tasks and models, significantly reducing overfitting and improving generalization in few-shot scenarios.

## 1 Introduction

Adapting Large Language Models (LLMs) to new tasks in few-shot learning scenarios can be achieved through either fine-tuning or In-Context Learning (ICL) (Brown et al., 2020). Parameter-efficient fine-tuning methods, such as Low-Rank Adaptation (LoRA) (Hu et al., 2021), which optimizes a subset of the model's parameters, and Prompt Tuning (PT) (Lester et al., 2021), which optimizes a small set of learnable tokens prepended to the input, aim to achieve task-specific performance with minimal computational overhead. In contrast, ICL eliminates the need for parameter updates by incorporating training examples directly into the input context, offering a training-free alternative that leverages the model's pre-trained knowledge without modifying its underlying parameters. Despite their effectiveness, determining the optimal approach for varying dataset sizes remains an ongoing challenge.

In few-shot scenarios with limited data, ICL has shown greater effectiveness; however, as the dataset size increases, optimization-based methods like LoRA and PT become preferable. This trend has been observed in prior studies (Mosbach et al., 2023; Min et al., 2022) and is further supported by our experimental results (fig. 6), which demonstrate that while ICL excels in low-data regimes, its advantage diminishes as more data becomes available. Although promising, the limitations of optimization-based methods in low-data scenarios require further exploration.

In this work, we identify overfitting as the main limitation of optimization-based methods in few-shot scenarios, as illustrated in fig. 1. To overcome this, we introduce Context-Aware Prompt Tuning (CPT), which extends ICL with a carefully designed optimization strategy that mitigates overfitting. CPT builds on ICL's inherent robustness by adding an optimization step inspired by Prompt Tuning (Lester et al., 2021), which updates the context prompt embeddings. However, a naive implementation of this optimization can still result in overfitting. To address this, we constrain the allowable changes per token, ensuring the optimized embeddings remain close to their original context. This regularization not only helps prevent overfitting but also maintains interpretability, as updated tokens retain semantic proximity to the initial samples. Additionally, CPT incorporates contextual information directly into its loss function, leveraging richer supervision

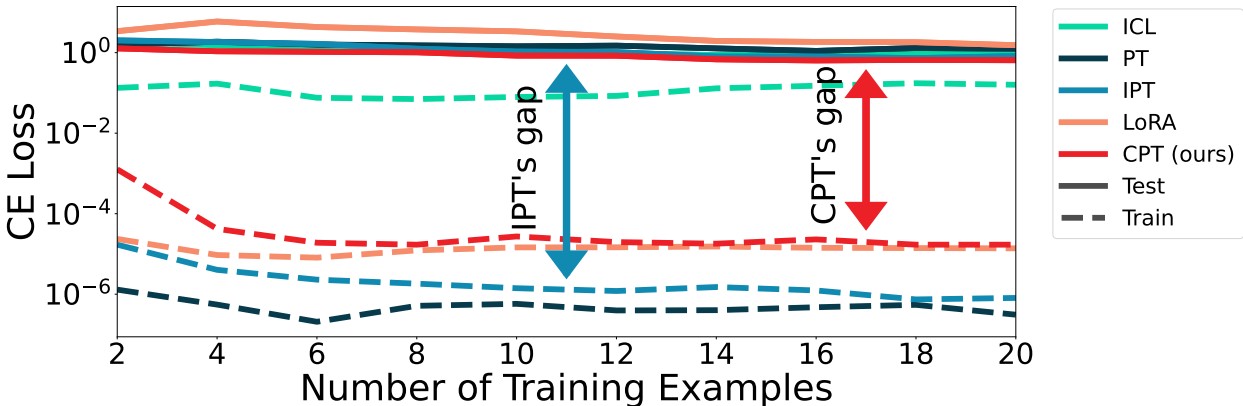

Figure 1: **Overfitting Across Few-Shot Methods** Train-test loss gap across methods and training set sizes using the GPT-J model on the DBpedia dataset. For each model, there are two loss graphs: one for train loss (dotted line) and one for test loss (solid line). CPT performs better in mitigating overfitting compared to optimization-based methods. Despite a relatively higher training loss, CPT achieves the lowest test loss.

signals from all embedded examples to further enhance generalization. Finally, CPT employs a loss weighting scheme based on recency bias (Zhao et al., 2021), encouraging the model to prioritize more recent—and thus typically more relevant—examples during optimization.

We rigorously evaluate CPT across various classification and question answering tasks and multiple model architectures, conducting extensive ablation studies to validate each design choice. To ensure robustness, we utilize diverse templates and random seeds—an essential consideration given ICL's known sensitivity to the selection and formatting of training examples Sun et al. (2023); Zhao et al. (2021). Our results consistently demonstrate CPT's superior performance over existing baselines across a wide range of scenarios.

To summarize, our key contributions are:

- We identify overfitting as the primary limitation of optimization-based methods in few-shot scenarios and empirically demonstrate its negative impact on performance.
- We propose CPT, a novel method that enhances ICL with a targeted optimization strategy designed explicitly to mitigate overfitting.
- We achieve state-of-the-art results across several classification and question answering benchmarks, accompanied by extensive ablation studies validating each component of CPT.

## 2 Related Work

**Fine-Tuning** Fine-tuning is a popular and effective method for adjusting LLMs to specific tasks. Standard fine-tuning (Radford et al., 2019; Brown et al., 2020; Howard & Ruder, 2018; Liu et al., 2019; Lan et al., 2019; Raffel et al., 2020; Sun et al., 2019) retrains the model with new data. However, a key disadvantage is the large number of parameters that must be stored.

**Efficient Fine-Tuning** To alleviate the computational burden of fine-tuning, Adapter-BERT (Houlsby et al., 2019) proposes training only the adapter layers inserted into the model, while BitFit (Zaken et al., 2021) focuses on fine-tuning just the bias terms. Delta Tuning (Ding et al., 2022) explores parameter-efficient methods that adjust only a small portion of a model's parameters. Low-Rank Adaptation methods (LoRA) (Hu et al., 2021) introduces a novel low-rank adaptation technique, where additional low-rank matrices are added to the weights during training. This allows the model to to train only these matrices, reducing the number of trainable parameters significantly. VERA (Kopiczko et al., 2023) builds on LoRA by incorporating adaptive learning rates. Compacter Karimi Mahabadi et al. (2021) leverages hypercomplex

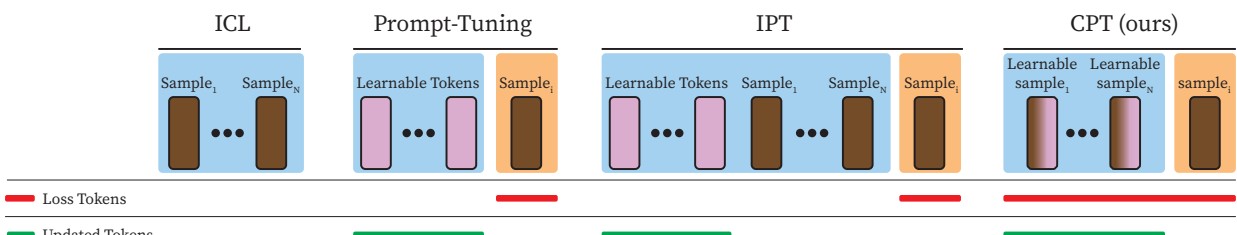

Figure 2: **Comparison of Few-Shot Methods** We highlight the key differences between *CPT* and the baseline methods, focusing on *ICL*, *PT*, and *IPT*. Each method includes two token types: prefix tokens (blue background) and loss tokens (orange background). The tokens are categorized into *Learnable Tokens* (pink) and *Sample Tokens* (brown), which remain fixed during training. A red line beneath the tokens indicates those used for loss calculation, while a green line marks those updated during training. *CPT* introduces *Learnable Sample* tokens, shown in a brown-pink color, initialized with training samples and progressively refined during optimization.

layers, and LoRA-Pro (Wang & Liang, 2024) further refines optimization. Despite these advancements, large models like GPT-3, which contain $175B$ parameters, require updating millions of parameters, such as 17.5M for LoRA.

**Prompt Tuning (PT)** Unlike fine-tuning methods, PT reduces the number of trainable parameters by introducing learnable tokens optimized while keeping the model's weights frozen. Lester et al. (2021) propose appending continuous prompts to the input and optimizing them, while P-tuning (Liu et al., 2023) and Prefix Tuning (Li & Liang, 2021) extend this concept by incorporating learnable tokens at intermediate layers. More recently, Wang et al. (2023) introduced the idea of training a single prompt to be shared across multiple tasks. Although these methods significantly reduce the number of trainable parameters, they face challenges in few-shot learning Gu et al. (2021) and provide limited interpretability for the learned continuous tokens (Ghosal et al., 2024; Khashabi et al., 2021; Deng et al., 2022).

**In-Context Learning (ICL)** In contrast to earlier methods, ICL (Brown et al., 2020) avoids optimization entirely. Instead, it concatenates task-specific examples before the input, allowing the model to learn a new task purely through observation, leveraging its pre-trained knowledge. Despite its advantages, ICL has limitations, often underperforming compared to optimization-based methods (Liu et al., 2022; Peng et al., 2023; Sun et al., 2023).

**Instruction Prompt Tuning (IPT)** IPT (Singhal et al., 2022) combines key elements of PT and ICL, utilizing learnable tokens that are optimized during training alongside static context tokens, similar to ICL. The concept of using both soft and hard prompts was previously introduced by PPT (Gu et al., 2021) and PTR (Han et al., 2022). Yet, IPT has struggled to consistently surpass PT in performance (Sun et al., 2023). While our method shares similarities with IPT, we focus on optimizing context tokens without introducing additional learnable tokens, and we are also leveraging context labels in the process. Another key difference lies in the optimization process, where our loss includes a regularization term, and we employ projected gradient descent to ensure the output stays close to the user-supplied reliable input.

## 3 Our Method

### 3.1 Overfitting in Few-Shot Learning

In few-shot learning scenarios with limited data, the risk of overfitting in optimization-based methods is closely tied to the number of trainable parameters they introduce. For example, when working with LLaMA 3 8B, methods such as full fine-tuning, LoRA, and PT involve updating approximately 8B, 4.2M, and 32K parameters, respectively. This demonstrates that even the most parameter-efficient optimization-based

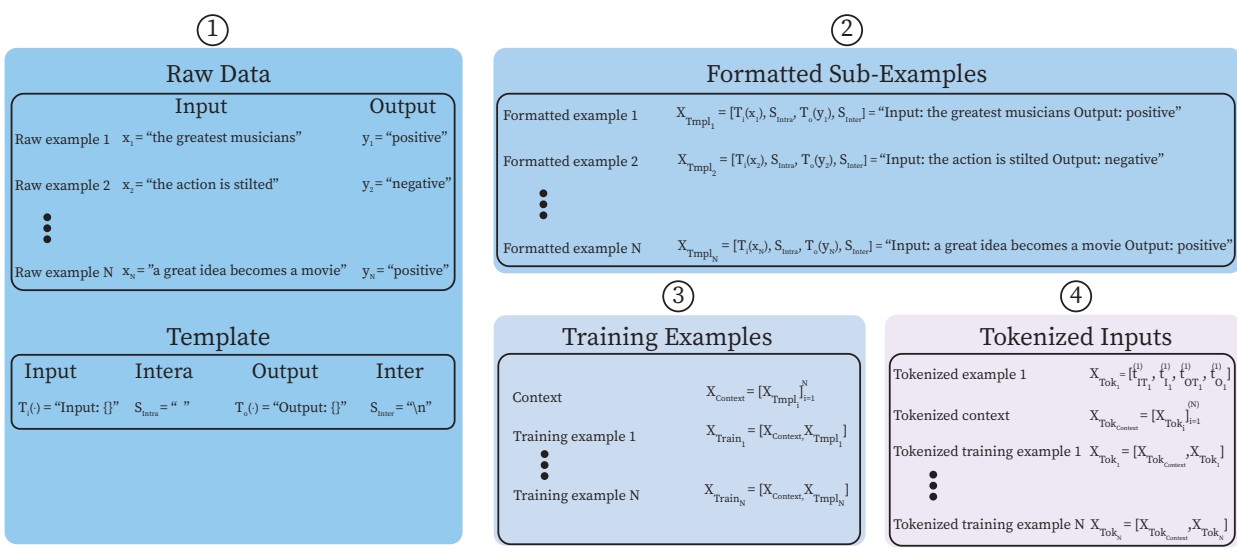

Figure 3: **Data Preparation Pipeline. (1)** We start with $N$ raw examples $(x_i, y_i)$ and a template. **(2)** Each example is formatted using $T_i(\cdot), S_{\text{Intra}}, T_o(\cdot), S_{\text{Inter}}$, producing $X_{\text{Tmpl}_i}$. **(3)** We concatenate all $X_{\text{Tmpl}_i}$ to form $X_{\text{Context}}$, then append a chosen $X_{\text{Tmpl}_i}$ to build $X_{\text{Train}_i}$. **(4)** Finally, we tokenize $X_{\text{Train}_i}$ into discrete tokens for the LLM.

methods still require training a significant number of parameters, which can pose challenges when data is limited.

Our analysis, presented in fig. 1, confirms that all optimization-based methods, including LoRA and PT, exhibit a train-test loss gap, highlighting overfitting in low-data regimes where the number of examples ranges from 2 to 20. Despite variations in the number of trainable parameters, these methods struggle to generalize effectively when data is limited. These findings emphasize the need for approaches like CPT, which is carefully designed to balance flexibility and generalization, effectively mitigating overfitting challenges in few-shot learning scenarios.

### 3.2 Input Preparation

We follow a four-step procedure to convert our $N$ raw input-output examples into the sequences required by the LLM, as illustrated in fig. 3:

**(1) Raw Data.** We begin with $N$ raw examples, each comprising an instruction $x_i$ and label $y_i$, along with a template $T_i(\cdot), S_{\text{Intra}}, T_o(\cdot), S_{\text{Inter}}$, specifying how $x_i$ and $y_i$ should be embedded.

**(2) Formatted Sub-Examples.** Applying the template to $(x_i, y_i)$ yields a formatted sequence $X_{\text{Tmpl}_i} = \left[ T_i(x_i), S_{\text{Intra}}, T_o(y_i), S_{\text{Inter}} \right]$. We do this for all $N$ examples, producing $X_{\text{Tmpl}_1}, \ldots, X_{\text{Tmpl}_N}$.

**(3) Training Examples.** We then concatenate *all* $X_{\text{Tmpl}_i}$ to form the *context*, $X_{\text{Context}} = [X_{\text{Tmpl}_1}, \ldots, X_{\text{Tmpl}_N}]$. Next, we append the $i$-th formatted sub-example $X_{\text{Tmpl}_i}$ to $X_{\text{Context}}$, yielding $X_{\text{Train}_i} = \left[ X_{\text{Context}}, X_{\text{Tmpl}_i} \right]$. Each $X_{\text{Train}_i}$ thus contains $N + 1$ sub-examples: $N$ in the context plus one appended "training" sub-example.

**(4) Tokenized Inputs.** Finally, we tokenize $X_{\text{Train}_i}$ into discrete tokens. Let $t_{\text{IT}_i}^{(k)}, t_{\text{I}_i}^{(k)}, t_{\text{OT}_i}^{(k)}, t_{\text{O}_i}^{(k)}$ represent input template, input, output-template and output tokens, where $k$ represents the sub-example. These tokenized inputs are then fed into the *frozen* LLM during training. See Appendix G for a concrete example.

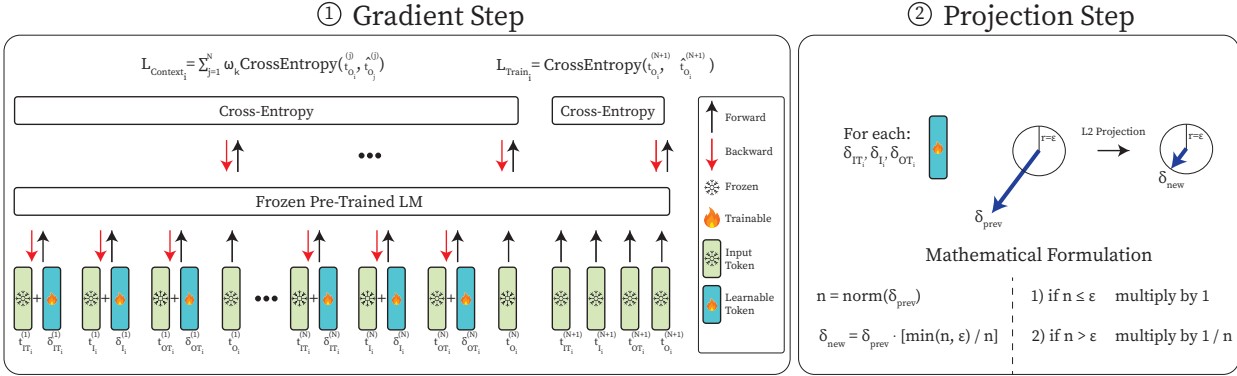

Figure 4: **Optimization Process in CPT.** This figure illustrates the two-step optimization procedure used in CPT. In the gradient step, gradients are used to update only the $\delta$ variables, while label tokens remain fixed. In the projection step, each vector $\delta$ is projected back onto the $\ell_2$ ball of radius $\epsilon$. This regularization prevents overfitting and preserves the interpretability of the original context examples.

## 3.3 Optimization

We now describe the optimization process, which combines the strengths of in-context learning (ICL) with the flexibility of gradient-based refinement. We begin by introducing our loss function in section 3.3.1, followed by a detailed explanation of the controlled token embedding optimization procedure in section 3.3.2. Each optimization iteration consists of two phases: a gradient step, followed by a projection step. Both the loss formulation and the update procedure are carefully designed to mitigate overfitting in few-shot settings.

### 3.3.1 Loss Design

We introduce a novel loss function for each training example $X_{\text{Train}_i}$ that combines the standard loss, denoted as $L_{\text{Train}_i}$, with an additional loss over the context sub-example labels from $X_{\text{Context}}$, denoted as $L_{\text{Context}_i}$. The context loss compares the model's predicted tokens $\hat{t}_{O_i}^{(k)}$ to the ground truth tokens $t_{O_i}^{(k)}$ for all $k \in [1, N]$, as illustrated in eq. (1).

$$L_{\text{Context}_i} = \sum_{k=1}^{N} \omega_k \cdot \text{CrossEntropy}(\hat{t}_{O_i}^{(k)}, t_{O_i}^{(k)}) \tag{1}$$

In addition, we apply the standard loss to the last sub-example in $X_{\text{Train}_i}$—referred to as the training sub-example—as defined in eq. (2).

$$L_{\text{Train}_i} = \text{CrossEntropy}(\hat{t}_{O_i}^{(N+1)}, t_{O_i}^{(N+1)}) \tag{2}$$

Lastly, we sum both losses to create the final loss $L_i = L_{\text{Context}_i} + L_{\text{Train}_i}$, where $L_{\text{Context}_i}$ can be thought of as a regularization for the standard loss $L_{\text{Train}_i}$.

As explained in section 3.2, each training example $X_{\text{Train}_i}$ contains $N + 1$ sub-labels, from $N$ sub-examples in the context and one training sub-example. However, not all sub-examples should be weighted equally. In particular, the last sub-example is most important, as it occupies the same position as test-time examples. This positional effect has been studied in prior work, which found that examples closer to the end of the context disproportionately influence model predictions (Zhao et al., 2021). Motivated by these findings, we apply exponential loss weight decay starting from the end of the context and decaying toward the beginning, while keeping $L_{\text{Train}_i}$ unchanged. Formally, each sub-example $k$ is weighted by $\omega_k = \gamma^j$, where $j = N + 1 - k$.

For example, the last sub-example is multiplied by $\gamma^1$, the second-to-last by $\gamma^2$, and so on. This weighting is illustrated in eq. (1).

### 3.3.2 Controlled Token Embedding Optimization

The context part of $X_{\text{Train}_i}$ serves a dual purpose: it is used both in the loss computation and as a set of parameters to be updated for the new task. In this section, we describe how we carefully control updates to the context to preserve its critical role in the loss, prevent overfitting, and enable effective learning. An illustration of the optimization process is shown in fig. 4.

Since the context serves two roles, a conflict arises: the portion that contributes to the loss contains ground truth labels, which must remain unchanged. To resolve this, we leave the label tokens untouched during optimization and update only the remaining tokens in the context. This is visualized in fig. 4, where the target tokens are excluded from the update process, $t_{O_i}^{(k)}$ for $k \in [1, N]$.

While this resolves the issue of label integrity, unconstrained updates to the remaining context tokens can still lead to overfitting, especially in few-shot scenarios where training data is limited. To mitigate this, we introduce a regularization strategy that limits the magnitude of updates applied to the token embeddings. Specifically, each token we aim to update is associated with a vector $\delta$ of the same dimensionality, representing its deviation from the original embedding. For each sub-example $k \in [1, N]$, we maintain a separate $\delta$ vector for each token type being updated: $\delta_{\text{IT}_i}^{(k)}$, $\delta_{\text{I}_i}^{(k)}$, and $\delta_{\text{OT}_i}^{(k)}$.

Each optimization iteration consists of two steps. First, we perform a gradient step, updating the $\delta$ vectors based on the computed gradients. Then, we perform a projection step, where each $\delta$ is projected back into an $\ell_2$ ball of radius $\epsilon$. This technique, known as Projected Gradient Descent (PGD), is widely used in the adversarial attacks literature (Blau et al., 2022; 2023; Carlini & Wagner, 2017; Athalye et al., 2018; Madry et al., 2017; Gowal et al., 2020) to enforce bounded perturbations.

Beyond preventing overfitting, constraining updates to the context embeddings has the additional benefit of maintaining their interpretability. Since the context is initialized with user-provided examples, keeping the optimized tokens close to their original representations ensures that the learned context remains meaningful—akin to in-context learning (ICL). This resemblance enables the model to draw from relevant patterns and structure present in the original examples, without excessive modification. As $\epsilon$ approaches zero, our method converges to standard ICL, which is known for its robustness and minimal overfitting. Thus, our controlled optimization not only stabilizes learning but also enhances transparency and faithfulness to the intended prompt structure.

## 4 Experimental Setup

In this section, we provide details regarding the datasets, models, baselines, and evaluation used in our experiments. Implementation details are provided in appendix F.

### 4.1 Datasets

In this work, we focus on a classification task and select a variety of datasets to ensure robust conclusions across different task types. We include SST-2 (Socher et al., 2013) for sentiment analysis, AG News (Zhang et al., 2015b) for news classification, DBpedia (Zhang et al., 2015a) for ontology classification, and TREC (Li & Roth, 2002) for question classification, more details are provided in appendix D. For question answering tasks, we use for common datasets: PIQA Bisk et al. (2020), BoolQ Clark & Gardner (2019) and CommonsenseQA Talmor et al. (2019).

### 4.2 Models

We use models of varying sizes and quality to ensure robust evaluation and conclusions. For the relatively small model, we use BLOOM1.7B (Scao et al., 2022), while for larger models, we opt for GPT-J6B(Wang

| Dataset | Method | BLOOM 1.7B | | | GPT-J 6B | | | LLaMA3 8B | | |
|---|---|---|---|---|---|---|---|---|---|---|
| | | 2 | 4 | 6 | 2 | 4 | 6 | 2 | 4 | 6 |
| SST-2 | Prefix | 47.80(±00.5) | 47.33(±03.1) | 49.00(±03.1) | 52.23(±00.8) | 52.50(±02.3) | 52.87(±05.8) | – | – | – |
| | ICL | 50.53(±04.3) | 60.83(±12.6) | 61.87(±14.9) | 50.57(±05.5) | 67.47(±14.1) | 77.47(±13.1) | 76.43(±13.2) | 80.63(±15.7) | 83.10(±13.7) |
| | PT† | 64.97(±07.6) | 65.07(±08.2) | 65.07(±08.1) | 57.10(±06.8) | 52.93(±07.8) | 55.70(±09.5) | 72.97(±16.6) | 73.47(±17.1) | 84.57(±12.7) |
| | PT | 56.03(±08.6) | 56.90(±08.9) | 58.33(±08.7) | 64.07(±07.7) | 64.37(±07.0) | 64.60(±07.4) | 64.27(±06.4) | 65.70(±06.8) | 67.03(±07.0) |
| | IPT† | 58.50(±12.3) | 61.83(±14.1) | 62.80(±15.3) | 51.50(±05.8) | **83.20**(±11.1) | 84.80(±12.3) | 86.90(±08.7) | 88.03(±12.5) | 94.40(±02.7) |
| | IPT | 48.50(±02.0) | 58.80(±11.6) | 61.87(±14.8) | 48.13(±00.7) | 82.27(±13.0) | 87.17(±07.2) | 57.20(±13.7) | 87.40(±10.6) | 90.43(±12.0) |
| | LoRA | **66.40**(±06.9) | 66.93(±06.5) | 66.90(±06.4) | **69.80**(±09.5) | 71.53(±10.2) | 73.17(±10.0) | 68.73(±11.4) | 71.27(±15.0) | 83.97(±12.1) |
| | CPT† | 59.53(±12.3) | **72.40**(±13.8) | **74.83**(±15.4) | 52.53(±07.6) | 82.03(±11.1) | **88.07**(±07.0) | **92.73**(±05.0) | 95.07(±04.1) | 96.40(±02.0) |
| | CPT | 50.77(±07.7) | 70.70(±12.0) | 74.10(±12.9) | 50.53(±05.5) | 82.90(±12.7) | 88.03(±10.7) | 83.83(±13.2) | **96.30**(±01.6) | **96.50**(±01.3) |
| AG News | Prefix | 24.87(±01.7) | 25.35(±05.8) | 26.02(±05.3) | 32.32(±05.9) | 33.33(±06.2) | 46.08(±12.7) | – | – | – |
| | ICL | 35.12(±10.5) | 34.28(±11.7) | 42.48(±12.2) | 66.73(±10.0) | 62.38(±13.3) | 69.57(±10.4) | 79.38(±08.8) | 82.32(±03.2) | 85.27(±03.1) |
| | PT† | 28.67(±05.2) | 30.73(±06.0) | 41.17(±10.9) | 37.85(±16.2) | 44.85(±13.7) | 62.92(±13.6) | 59.60(±11.9) | 57.02(±10.9) | 68.02(±10.5) |
| | PT | 33.57(±07.9) | 36.98(±08.4) | **56.08**(±09.9) | 56.85(±12.3) | 56.13(±10.8) | 75.10(±07.0) | 69.32(±15.0) | 67.92(±15.4) | 69.33(±12.4) |
| | IPT† | 36.95(±11.7) | 31.90(±07.5) | 42.93(±15.0) | 67.02(±11.0) | 63.00(±07.1) | 74.85(±07.6) | 82.93(±02.7) | **84.45**(±03.5) | 85.08(±03.4) |
| | IPT | 38.77(±12.0) | 38.20(±10.6) | 47.78(±12.0) | 66.02(±11.1) | 63.92(±08.8) | 74.00(±07.4) | 80.52(±03.6) | 76.30(±08.1) | 80.98(±05.2) |
| | LoRA | 29.50(±03.2) | 30.80(±06.8) | 33.98(±04.6) | 56.12(±09.0) | 56.03(±09.4) | 72.55(±08.8) | 70.62(±15.0) | 74.97(±15.4) | 73.70(±12.4) |
| | CPT† | 33.68(±07.2) | 33.13(±06.0) | 41.10(±11.9) | 71.35(±09.6) | **68.73**(±09.0) | 75.68(±09.2) | 83.17(±03.1) | 84.28(±02.9) | 84.67(±03.4) |
| | CPT | **40.85**(±12.8) | **44.48**(±12.2) | 50.40(±11.3) | **74.80**(±08.6) | 68.62(±09.1) | **76.22**(±07.2) | 83.78(±03.3) | 81.92(±04.3) | **85.43**(±02.9) |
| DBpedia | Prefix | 19.76(±03.5) | 19.74(±06.4) | 23.65(±08.7) | 13.25(±02.3) | 16.43(±04.5) | 24.94(±07.9) | – | – | – |
| | ICL | 48.20(±24.1) | 51.40(±25.8) | 55.17(±23.9) | 50.87(±16.6) | 62.46(±15.7) | 70.76(±06.7) | 71.66(±07.7) | 72.44(±06.8) | 79.93(±04.2) |
| | PT† | 24.90(±15.6) | 26.32(±09.7) | 34.75(±07.1) | 21.01(±10.3) | 22.12(±06.3) | 37.44(±06.3) | 55.30(±19.5) | 57.21(±15.7) | 66.26(±15.4) |
| | PT | 46.71(±11.2) | 41.94(±10.8) | 45.93(±13.0) | 23.39(±09.9) | 29.69(±11.3) | 40.53(±08.9) | 55.81(±11.9) | 52.72(±15.7) | 55.02(±13.5) |
| | IPT† | 33.28(±25.6) | 40.36(±24.3) | 45.85(±27.3) | 47.10(±16.4) | 67.60(±11.4) | 75.09(±06.8) | 81.10(±05.3) | 87.69(±04.5) | 92.06(±04.5) |
| | IPT | 48.09(±26.2) | 54.60(±25.0) | 70.57(±07.6) | 52.86(±12.2) | 67.27(±09.6) | 70.73(±05.4) | 72.92(±09.7) | 76.11(±06.0) | 78.44(±05.6) |
| | LoRA | 43.30(±11.4) | 41.13(±11.6) | 41.18(±11.7) | 30.15(±11.6) | 28.02(±13.0) | 41.50(±09.8) | 54.24(±13.1) | 59.50(±14.3) | 63.21(±13.7) |
| | CPT† | 33.80(±23.2) | 48.13(±12.0) | 51.18(±22.1) | 53.20(±15.6) | **77.30**(±06.5) | **81.00**(±06.0) | **84.23**(±06.2) | **90.33**(±03.8) | **93.08**(±02.3) |
| | CPT | **58.85**(±15.5) | **65.78**(±11.8) | **73.55**(±04.7) | **68.29**(±10.9) | 75.07(±05.0) | 77.65(±03.9) | 77.38(±06.1) | 78.49(±04.3) | 82.42(±04.3) |
| TREC | Prefix | 19.10(±06.7) | 24.49(±06.4) | 29.92(±07.0) | 30.76(±03.1) | 30.04(±05.9) | 27.87(±03.7) | – | – | – |
| | ICL | 33.54(±11.0) | 33.33(±10.5) | 28.53(±13.8) | 28.94(±08.9) | 35.14(±11.0) | 32.49(±12.6) | 35.32(±08.6) | 42.48(±14.2) | 40.34(±13.2) |
| | PT† | 30.91(±05.9) | 33.70(±07.5) | 39.31(±11.2) | 29.02(±05.5) | 34.66(±06.2) | 43.89(±13.5) | 43.42(±09.5) | 48.81(±11.3) | 51.73(±10.3) |
| | PT | 32.18(±03.8) | 32.26(±08.2) | 35.69(±11.2) | 31.16(±04.0) | 32.79(±08.1) | 37.86(±09.7) | 32.77(±05.0) | 33.98(±04.5) | 33.83(±03.8) |
| | IPT† | 27.83(±05.5) | 36.64(±10.1) | 42.92(±16.8) | 31.04(±06.8) | 43.12(±07.7) | 43.09(±14.0) | 51.72(±13.6) | 62.14(±09.7) | 65.13(±07.2) |
| | IPT | 32.37(±10.5) | 36.59(±06.1) | 42.60(±14.1) | 29.59(±09.7) | 38.90(±09.3) | 40.38(±13.0) | 36.94(±12.1) | 45.62(±11.5) | 52.08(±14.0) |
| | LoRA | 34.07(±03.9) | 33.22(±04.0) | 33.50(±04.1) | 34.17(±02.5) | 33.73(±03.6) | 37.63(±11.9) | 31.21(±03.3) | 33.21(±03.5) | 36.36(±16.5) |
| | CPT† | 29.72(±08.0) | 35.64(±07.7) | **45.38**(±09.9) | 33.39(±08.5) | 44.20(±12.9) | **45.83**(±11.2) | **57.26**(±13.1) | **67.00**(±09.8) | **69.29**(±05.0) |
| | CPT | **35.68**(±09.1) | **41.79**(±07.9) | 45.16(±12.9) | **35.37**(±07.4) | **44.66**(±08.7) | 42.71(±08.6) | 45.12(±16.8) | 57.54(±07.4) | 60.18(±07.9) |

Table 1: **Baseline Comparisons** Mean accuracy of various methods and our CPT, across several models and datasets. Evaluations are conducted using 2, 4, and 6 shots.

& Komatsuzaki, 2021) and LLaMA3 8B(AI@Meta, 2024). The GPT-J model is noted for its robust performance, while LLaMA3 is currently among the leading models in the field.

## 4.3 Baselines

We compare our method to several groups of few-shot learning techniques. In the first group, we include LoRA (Hu et al., 2021), one of the leading efficient fine-tuning methods. Additionally, we compare against several prompt-tuning approaches, including Prompt Tuning (PT) (Lester et al., 2021), Prefix Tuning (Li & Liang, 2021), and Instruction Prompt Tuning (IPT) (Singhal et al., 2022). Finally, we compare our method to In-Context Learning (ICL) (Brown et al., 2020).

For some of the few-shot methods, we introduce an alternative version that incorporates instructions, as indicated in table 1 with a †. Instead of initializing the learnable tokens randomly, we initialize them with instructions specified in appendix C. We apply instructions to PT, IPT, and our method, reporting results for both random and instruction-based prompt initialization. An example illustrating how inputs are constructed with and without †is provided in appendix G.

## 4.4 Evaluation

We evaluate each model and dataset using three different numbers of training samples: 2, 4, and 6. For each configuration, the reported results are averaged accuracy over 30 experiments, consisting of 10 randomly sampled templates and 3 different random seeds, with the templates described in appendix E. By utilizing

randomized seeds, we ensure variation in the selection of training examples. This extensive setup is crucial for achieving a comprehensive and robust evaluation, especially given that these methods are known to be highly sensitive to the selection of training examples and templates (Voronov et al., 2024; Zhao et al., 2021). Further evaluation details can be found in appendix B.

# 5 Results

## 5.1 Main Results

In table 1, we compare performance on classification tasks and show that CPT convincingly performs better than the baselines in most cases, with particularly pronounced gains in harder tasks. Furthermore, CPT 's performance becomes more efficient and effective as the models grow stronger, such as with LLaMA3. In table 2, CPT likewise demonstrates strong performance on QA tasks.

**Performance on Challenging Tasks** CPT demonstrates improvements across various datasets, with more pronounced gains in tasks we define as harder based on two factors: the number of shots and the number of classes. As illustrated in table 1, task difficulty increases with fewer shots and more classes. For example, on the DBpedia dataset, which has 14 classes, decreasing the shots from 6 to 4 widens the performance gap between CPT and the baselines from (3, 6, 1) to (11, 10, 3) across the models: BLOOM, GPT-J, and LLaMA3.

**Decisive Advantage with Powerful Models** The strength of the model plays a significant role in performance. As the model becomes better, CPT's advantage becomes more pronounced across all datasets and shot settings. For instance, LLaMA3 consistently outperforms other baselines across all datasets, except in one case where results are comparable. With GPT-J, a slightly older model, the results are lower in two instances, with one comparable outcome, both on SST-2 , the easier task as previously discussed. When comparing with BLOOM , the weakest model in our comparison, we observe lower performance on two occasions, specifically on the two easier datasets.

| Dataset | Method | GPT-J 6B | | LLaMA3 8B | |
| --- | --- | --- | --- | --- | --- |
| | | 2 | 4 | 2 | 4 |
| BoolQ | ICL | 63.00(±01.0) | 63.74(±01.6) | 62.22(±00.1) | 62.17(±00.0) |
| | PT | 62.10(±00.0) | 62.10(±00.0) | 66.20(±07.5) | 68.72(±11.0) |
| | IPT | 63.26(±01.1) | 64.53(±02.2) | 64.28(±02.5) | 69.80(±07.5) |
| | LoRA | 62.66(±00.9) | 62.94(±00.7) | 70.01(±10.7) | 73.33(±10.9) |
| | CPT | **64.56**(±01.7) | **65.79**(±03.2) | **78.10**(±01.8) | **78.11**(±13.8) |
| CommonsenseQA | ICL | 21.12(±01.2) | 20.92(±01.6) | 69.89(±02.3) | 71.39(±04.6) |
| | PT | 20.51(±00.8) | 21.99(±02.1) | 62.97(±01.6) | 65.30(±00.8) |
| | IPT | 21.47(±01.4) | 21.36(±01.7) | 68.41(±02.7) | 69.80(±05.1) |
| | LoRA | 20.08(±00.7) | 19.78(±00.3) | 63.38(±00.2) | 63.79(±00.9) |
| | CPT | **22.59**(±01.9) | **22.59**(±02.6) | **71.93**(±01.3) | **73.27**(±02.9) |
| PIQA | ICL | 50.66(±02.9) | 49.39(±01.1) | 56.63(±07.1) | 57.80(±04.0) |
| | PT | 50.63(±00.7) | 50.59(±00.8) | 51.83(±05.2) | 52.09(±04.3) |
| | IPT | 50.80(±02.0) | 50.00(±03.8) | 58.66(±09.5) | 61.16(±08.6) |
| | LoRA | 50.51(±00.5) | 50.51(±00.8) | 66.24(±00.8) | 69.18(±01.1) |
| | CPT | **50.84**(±01.3) | **51.42**(±00.9) | **73.14**(±08.5) | **70.56**(±11.1) |

Table 2: **Performance comparison on QA tasks** under few-shot settings, comparing multiple baselines across diverse QA datasets, model architectures, and seeds.

**Performance on Question Answering Tasks** CPT consistently outperforms all baselines across the QA datasets, each of which poses distinct challenges. As shown in table 2, CPT's improvements are particularly pronounced with stronger models such as LLaMA3. These gains indicate that CPT not only mitigates overfitting but also enhances the model's ability to extract and generalize from contextual cues—an essential requirement in QA tasks. Furthermore, the low standard deviations demonstrate CPT's robustness to variations in both prompts and seeds, underscoring its reliability in sensitive few-shot QA settings.

## 5.2 Standard Deviation

Standard deviation (std) plays a crucial role in few-shot learning due to the sensitivity of these methods to both the training examples and the chosen template (Zhao et al., 2021; Voronov et al., 2024). In fig. 5, we present accuracy along with two types of std bars: black bars represent the mean std across different templates, while blue bars represent the mean std across different seeds. We demonstrate that CPT significantly improves accuracy across various models and datasets in a statistically significant manner. More information in appendix A.

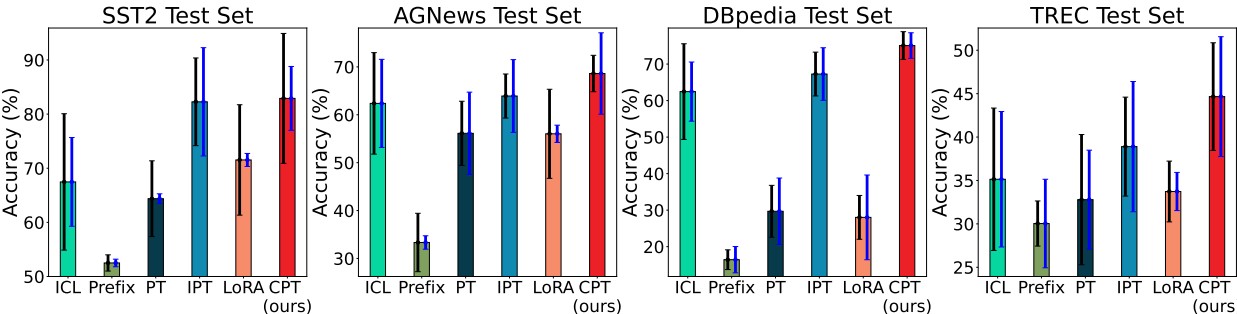

Figure 5: **Accuracy and Standard Deviation** Comparison of accuracy and standard deviation between CPT and baselines, evaluated with 4-shot on GPT-J model. The black bars represent the mean std across different templates, while the blue bars represent the mean std across different seeds.

Our method's standard deviation performs equivalently to other methods in most cases, while in certain cases, such as with DBpedia, CPT exhibits both higher accuracy and lower std, reinforcing its robustness in complex tasks. However, the sensitivity of our method does not follow a clear pattern across random seeds or templates. For instance, while randomness in templates and training examples has an equal influence on std in DBpedia and TREC, SST-2 shows a higher std for template randomness, and AG News is more sensitive to variations in training examples.

## 5.3 Ablations

Our ablation studies aim to dissect the contributions of individual components in CPT, highlighting the elements that drive its performance improvements across few-shot learning tasks, as shown in table 3. As shown, the loss design and the projections are the most important component of out method. Further ablation experiments can be found in appendix I.

**Loss Design** Different options for the loss design are specified under "Loss Tokens", with three configurations: using only the training label, using the training label plus one random context label, and using the training label plus all context labels. The latter outperforms the training-only configuration by $11\%, 12\%, 10\%$ for 2, 4 and 6 shots.

**Effect of Projection Magnitude** The ablation study on projection magnitude is specified under "Input $\epsilon$" and "Format $\epsilon$", which define the allowable deviation from the original values for input tokens and format tokens, respectively. The results demonstrate that both excessively small changes (leading to convergence toward ICL) and overly large norms (failing to limit overfitting) are suboptimal, emphasizing the importance of selecting an appropriate projection magnitude.

**Loss Weighting** We evaluated the impact of different loss weighting strategies and propose three options: (1) *Mean*, which applies uniform weighting across all labels; (2) *Equal*, which assigns equal weight to the training label loss and the context label losses, with an optional scaling factor applied to the training loss (*e.g.*, 1, 10); and (3) *Decay*, which exponentially reduces the weight of context labels further from the training example, with the decay factor specified (*e.g.*, 0.99, 0.95, 0.5).

**Projection Type: Token-wise vs. All-Tokens** We evaluated the "All-Tokens" projection approach, which applies the projection to the entire context collectively rather than processing it token-by-token. Our results indicate that the token-wise approach is preferable, as it provides stronger regularization by limiting each token individually rather than the context as a whole, resulting in better performance.

**Updated Tokens** Under "Updated Tokens", we explored modifying only specific parts of the context to determine if certain components are more critical for updates than others. Our results indicate that allowing changes to both the input and format tokens yields better performance, provided these changes are constrained using the projection limitation.

| Loss Tokens | Loss Weighting | Projection Type | Input $\epsilon$ | Format $\epsilon$ | Updated Tokens | Mask Training Example | Number of Training Examples | | |
| --- | --- | --- | --- | --- | --- | --- | --- | --- | --- |
| | | | | | | | 2 | 4 | 6 |
| Train Example | | | | | | | 58.09(±12.2) | 61.54(±15.3) | 66.69(±08.3) |
| Train Example & 1 Random | Decay 0.95 | Token-Wise | 0.1 | 0.1 | Input & Format | ✗ | 69.48(±07.2) | 72.08(±07.1) | 76.80(±04.4) |
| Train Example & All Context | | | | | | | 69.54(±09.9) | 73.03(±07.5) | 76.58(±04.1) |
| | Mean | | | | | | 69.62(±10.1) | 72.91(±07.4) | 76.49(±04.4) |
| | Equal 1 | | | | | | 69.07(±09.1) | 72.82(±07.9) | 76.23(±04.0) |
| Train Example & All Context | Equal 10 | Token-Wise | 0.1 | 0.1 | Input & Format | ✗ | 69.35(±11.3) | 71.01(±08.3) | 75.11(±04.6) |
| | Decay 0.99 | | | | | | 69.59(±10.0) | 72.97(±07.4) | 76.43(±04.4) |
| | Decay 0.95 | | | | | | 69.54(±09.9) | 73.03(±07.5) | 76.58(±04.1) |
| | Decay 0.5 | | | | | | 69.60(±08.6) | 72.39(±07.5) | 76.44(±03.5) |
| | | | 0.001 | - | | | 51.52(±16.3) | 63.41(±14.7) | 71.50(±05.6) |
| Train Example & All Context | Decay 0.95 | All-Tokens | 0.01 | - | Input & Format | ✗ | 56.37(±12.9) | 68.12(±08.8) | 73.66(±04.5) |
| | | | 0.1 | - | | | 69.51(±09.7) | 72.64(±08.1) | 76.06(±04.2) |
| | | | 1.0 | - | | | 63.11(±14.3) | 64.78(±10.9) | 71.94(±05.7) |
| | | | 0.01 | 0.1 | | | 65.61(±09.9) | 70.12(±08.5) | 75.63(±04.2) |
| | | | 0.1 | 0.1 | | | 69.54(±09.9) | 73.03(±07.5) | 76.58(±04.1) |
| Train Example & All Context | Decay 0.95 | Token-Wise | 1.0 | 0.1 | Input & Format | ✗ | 65.29(±11.8) | 66.30(±11.0) | 73.63(±06.2) |
| | | | 0.1 | 0.01 | | | 69.53(±10.5) | 73.55(±06.1) | 76.55(±04.1) |
| | | | 0.1 | 1.0 | | | 68.27(±10.4) | 71.91(±07.1) | 68.27(±04.2) |
| | | | | | Input | | 69.47(±10.4) | 74.13(±05.7) | 76.63(±04.0) |
| Train Example & All Context | Decay 0.95 | Token-Wise | 0.1 | 0.1 | Masks | ✗ | 63.74(±11.1) | 69.21(±09.5) | 74.91(±04.2) |
| | | | | | Input & Format | | 69.54(±09.9) | 73.03(±07.5) | 76.58(±04.1) |
| Train Example & All Context | Decay 0.95 | Token-Wise | 0.1 | 0.1 | Input & Format | ✓ | 67.55(±08.1) | 64.26(±10.0) | 68.58(±05.7) |

Table 3: **Ablation Study** We present the mean accuracy for various ablations using the GPT-J model and the DBpedia dataset, including loss tokens (train example, random, or all context), loss weighting (decay and mean), projection type (token-wise or all-tokens), epsilon values for input and format, updated tokens (input, format, masks), and masking of the training example.

**Mask Training** We also experimented with "Mask Training," where the training example was masked from the context to prevent the model from simply copying the answer. In our setup, the training example appears both in the context (along with the correct answer) and as an additional concatenated example at the end. Masking the training example from the context and removing this duplication seemed like a plausible strategy to improve generalization. However, this approach did not lead to any performance improvements.

## 6 Discussions

In this work, we identify overfitting as the primary reason for the underperformance of optimization-based methods in few-shot learning scenarios, substantiated by empirical evidence. To address this challenge, we propose CPT , an optimization-based method that effectively mitigates overfitting. Our results demonstrate that CPT consistently outperforms existing baselines across diverse datasets, models, and experimental setups.

Beyond its direct contributions to few-shot learning, CPT highlights the critical importance of balancing optimization flexibility and regularization in data-scarce scenarios. The insights from this work can inspire the development of parameter-efficient, robust, and interpretable approaches for a range of machine learning challenges, including transfer learning, domain adaptation, and fine-tuning in resource-constrained environments.

**Limitation & Future Work** While CPT significantly improves performance over ICL, it incurs additional computational cost due to the iterative optimization of context embeddings , as presented in appendix K, and is therefore less suited for larger training sets or longer input sequences . Moreover, as shown in fig. 6 in appendix J, CPT shares a common limitation with ICL and IPT: since all training examples are embedded directly into the prompt, it is best suited for low-resource regimes where the total context length remains manageable. In contrast, traditional fine-tuning methods are more appropriate for larger training sets or longer input sequences.

Future work could explore more efficient optimization strategies to reduce computational overhead and improve scalability. Additionally, it would be valuable to investigate how CPT can be extended to other NLP settings.

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

| Dataset | Method | Model | | | | | | | | |
|---|---|---|---|---|---|---|---|---|---|---|
| | | BLOOM 1.7B | | | GPT-J 6B | | | LLaMA3 8B | | |
| | | 2 | 4 | 6 | 2 | 4 | 6 | 2 | 4 | 6 |
| SST-2 | Prefix Tuning | 00.5/00.4/00.1 | 03.1/02.7/01.9 | 03.1/02.6/02.1 | 00.8/00.6/00.2 | 02.3/01.5/00.7 | 05.8/04.3/03.9 | – | – | – |
| | ICL | 04.3/04.0/01.5 | 12.6/08.6/09.4 | 14.9/10.6/09.7 | 05.5/04.0/03.0 | 14.1/12.6/08.2 | 13.1/09.9/09.9 | 13.2/12.7/06.1 | 15.7/11.9/10.2 | 13.7/12.2/06.5 |
| | PT† | 07.6/07.6/00.4 | 08.2/08.2/00.7 | 08.1/08.1/00.6 | 06.8/06.5/02.3 | 07.8/06.9/04.3 | 09.5/09.0/04.9 | 16.6/16.5/01.0 | 17.1/17.1/01.6 | 12.7/10.9/05.7 |
| | PT | 08.6/08.5/01.7 | 08.9/08.6/02.4 | 08.7/08.4/02.5 | 07.7/07.6/01.3 | 07.0/07.0/00.9 | 07.4/07.4/01.0 | 06.4/06.1/03.1 | 06.8/06.6/02.8 | 07.0/06.5/03.7 |
| | IPT† | 12.3/12.3/03.4 | 14.1/11.4/09.8 | 15.3/10.1/12.6 | 05.8/05.1/02.5 | 11.1/08.0/08.3 | 12.3/12.3/02.0 | 08.7/06.8/05.6 | 12.5/11.2/07.3 | 02.7/02.6/01.6 |
| | IPT | 02.0/01.5/00.4 | 11.6/08.7/08.2 | 14.8/09.0/11.8 | 00.7/00.4/00.2 | 13.0/08.1/10.0 | 07.2/05.7/04.7 | 13.7/13.7/03.9 | 10.6/09.6/05.5 | 12.0/10.3/05.3 |
| | LoRA | 06.9/06.9/00.2 | 06.5/06.5/00.5 | 06.4/06.3/00.5 | 09.5/09.5/00.8 | 10.2/10.2/01.2 | 10.0/09.9/01.4 | 11.4/11.4/01.4 | 15.0/14.6/07.6 | 12.1/11.9/07.2 |
| | CPT† | 12.3/12.3/03.6 | 13.8/09.8/10.5 | 15.4/14.2/09.6 | 07.6/06.7/03.2 | 11.1/09.1/07.0 | 07.0/06.2/04.1 | 05.0/04.3/02.5 | 04.1/03.0/02.3 | 02.0/01.7/01.2 |
| | CPT | 07.7/05.3/02.9 | 12.0/11.1/06.9 | 12.9/10.5/09.7 | 05.5/03.9/02.9 | 12.7/12.0/05.9 | 10.7/08.9/05.1 | 13.2/10.9/08.5 | 01.6/01.5/01.0 | 01.3/01.2/01.0 |
| AG News | Prefix Tuning | 01.7/01.7/00.6 | 05.8/02.9/05.2 | 05.3/03.9/03.9 | 05.9/05.9/00.5 | 06.2/06.1/01.4 | 12.7/09.4/08.2 | – | – | – |
| | ICL | 10.5/06.8/08.7 | 11.7/10.1/06.4 | 12.2/11.1/06.0 | 10.0/08.9/05.2 | 13.3/10.6/09.2 | 10.4/09.4/05.0 | 08.8/03.2/08.1 | 03.2/03.0/02.3 | 03.1/02.7/02.4 |
| | PT† | 05.2/04.1/04.1 | 06.0/04.3/04.4 | 10.9/10.5/07.3 | 16.2/16.1/00.9 | 13.7/11.5/09.5 | 13.6/11.9/07.8 | 11.9/11.8/06.1 | 10.9/10.3/05.8 | 10.5/09.0/06.5 |
| | PT | 07.9/04.6/06.9 | 08.4/06.8/05.7 | 09.9/09.3/05.2 | 12.3/11.5/05.3 | 10.8/06.7/08.6 | 07.0/02.5/06.8 | 15.0/15.0/01.5 | 15.4/15.2/02.7 | 12.4/12.3/02.2 |
| | IPT† | 11.7/09.2/08.4 | 07.5/05.5/05.0 | 15.0/09.4/12.4 | 11.0/08.3/07.9 | 07.1/03.1/06.5 | 07.6/03.5/06.8 | 02.7/02.5/01.6 | 03.5/02.8/02.2 | 03.4/02.9/02.5 |
| | IPT | 12.0/08.6/08.9 | 10.6/09.1/07.2 | 12.0/09.4/07.9 | 11.1/10.0/06.3 | 08.8/04.6/07.6 | 07.4/04.6/05.7 | 03.6/03.1/01.8 | 08.1/03.2/07.6 | 05.2/02.2/04.7 |
| | LoRA | 03.2/03.2/00.5 | 06.8/03.3/06.1 | 04.6/04.4/02.7 | 09.0/09.0/01.3 | 09.4/09.3/01.8 | 08.8/05.3/07.3 | 03.1/02.5/02.2 | 02.9/02.6/02.4 | 03.4/01.8/03.2 |
| | CPT† | 07.2/05.7/05.4 | 06.0/03.4/04.8 | 11.9/08.5/09.2 | 09.6/07.2/07.1 | 09.0/03.6/08.5 | 09.2/04.4/08.1 | 03.1/02.5/02.2 | 02.9/02.6/02.4 | 03.4/01.8/03.2 |
| | CPT | 12.8/08.7/10.3 | 12.2/07.8/10.3 | 11.3/08.9/07.1 | 08.6/05.4/07.4 | 09.1/03.8/08.5 | 07.2/03.7/06.2 | 03.3/02.6/02.3 | 04.3/03.6/02.8 | 02.9/02.3/02.3 |
| DBpedia | Prefix Tuning | 03.5/03.4/01.9 | 06.4/03.1/05.7 | 08.7/03.6/08.1 | 02.3/02.3/01.7 | 04.5/02.7/03.6 | 07.9/04.9/06.2 | – | – | – |
| | ICL | 24.1/23.3/06.9 | 25.8/23.5/08.9 | 23.9/23.6/06.0 | 16.6/16.3/05.9 | 15.7/13.1/08.1 | 06.7/05.8/04.0 | 07.7/06.4/06.2 | 06.8/02.6/06.5 | 04.2/02.3/04.0 |
| | PT† | 15.6/08.5/13.2 | 09.7/09.6/01.7 | 07.1/04.6/05.8 | 10.3/10.3/00.9 | 06.3/05.8/04.2 | 06.3/05.8/04.2 | 19.5/17.3/11.0 | 15.7/12.7/09.0 | 15.4/13.7/08.2 |
| | PT | 11.2/11.1/04.2 | 10.8/10.8/01.3 | 13.0/12.4/05.9 | 09.9/08.2/06.4 | 11.3/07.1/09.1 | 08.9/05.0/07.5 | 11.9/11.7/04.2 | 15.7/15.3/02.9 | 13.5/13.4/01.5 |
| | IPT† | 25.6/21.2/14.7 | 24.3/22.6/08.9 | 27.3/26.2/08.5 | 16.4/15.5/05.7 | 11.0/09.7/06.2 | 06.8/05.3/05.2 | 05.3/04.3/03.0 | 04.5/03.8/02.7 | 04.5/04.1/01.9 |
| | IPT | 26.2/25.1/07.5 | 25.0/20.3/11.9 | 07.6/07.0/02.9 | 12.2/11.2/06.0 | 09.6/06.0/07.2 | 05.4/03.7/04.1 | 09.7/08.4/04.6 | 06.0/03.6/05.1 | 05.6/03.1/05.3 |
| | LoRA | 11.4/11.0/03.1 | 11.6/11.6/00.3 | 11.7/11.7/00.4 | 11.6/10.9/04.9 | 11.6/10.9/04.9 | 09.8/06.0/07.9 | 13.1/13.0/01.7 | 14.3/14.2/02.2 | 13.7/13.7/01.5 |
| | CPT† | 23.2/14.5/18.0 | 12.0/10.1/07.1 | 22.1/20.1/10.5 | 15.6/08.6/14.3 | 06.5/05.0/04.5 | 06.0/03.2/05.2 | 06.2/05.6/02.7 | 03.8/03.4/02.4 | 02.3/02.2/01.7 |
| | CPT | 15.5/13.4/06.5 | 11.8/08.8/07.1 | 04.7/04.0/02.5 | 10.9/08.2/05.5 | 05.0/03.8/03.5 | 03.9/03.0/03.0 | 06.1/05.0/04.8 | 04.3/03.5/03.0 | 04.3/02.6/03.5 |
| TREC | Prefix Tuning | 06.7/00.8/06.6 | 06.4/02.9/06.0 | 07.0/04.6/06.0 | 03.1/02.0/02.5 | 05.9/02.6/05.1 | 03.7/03.6/00.8 | – | – | – |
| | ICL | 11.0/07.2/08.2 | 10.5/08.2/06.8 | 13.8/09.0/09.1 | 08.9/05.9/06.8 | 11.0/08.2/07.8 | 12.6/08.3/09.4 | 08.6/05.6/06.3 | 14.2/07.9/12.0 | 13.2/08.3/10.7 |
| | PT† | 05.9/04.4/03.7 | 07.5/06.7/04.5 | 11.2/08.7/07.8 | 05.5/04.3/03.4 | 06.2/06.0/04.3 | 13.5/08.2/11.7 | 09.5/05.7/08.3 | 11.3/06.1/09.4 | 10.3/08.4/08.1 |
| | PT | 03.8/03.4/01.5 | 08.2/07.3/06.7 | 11.2/08.6/09.1 | 04.0/04.0/00.9 | 08.1/07.5/05.7 | 09.7/08.2/07.8 | 05.0/05.0/01.5 | 04.5/04.5/02.5 | 03.8/03.8/02.0 |
| | IPT† | 05.5/03.6/04.0 | 10.1/09.1/07.1 | 16.8/08.1/15.5 | 06.8/05.3/04.2 | 07.7/05.1/05.9 | 14.0/07.8/11.9 | 13.6/06.3/12.3 | 09.7/05.9/08.6 | 07.2/05.4/05.5 |
| | IPT | 10.5/07.1/07.8 | 06.1/05.9/05.0 | 14.1/07.4/12.8 | 09.7/05.4/08.0 | 09.3/05.7/07.5 | 13.0/03.8/12.5 | 12.1/08.5/07.8 | 11.5/09.1/08.0 | 14.0/05.4/13.3 |
| | LoRA | 03.9/03.9/01.0 | 04.0/04.0/00.3 | 04.1/04.1/00.4 | 02.5/02.5/00.4 | 03.6/03.5/02.2 | 11.9/07.2/10.4 | 03.3/03.3/01.0 | 14.3/14.2/02.2 | 16.5/08.1/15.4 |
| | CPT† | 08.0/05.8/05.4 | 07.7/06.9/06.3 | 09.9/07.0/07.9 | 08.5/05.7/06.4 | 12.9/08.3/10.6 | 11.2/08.2/09.0 | 13.1/06.9/11.6 | 09.8/03.7/09.2 | 05.0/04.1/03.4 |
| | CPT | 09.1/05.2/07.3 | 07.9/07.2/05.6 | 12.9/07.0/10.8 | 07.4/04.1/06.1 | 08.7/06.2/06.9 | 08.6/05.5/07.3 | 16.8/08.4/14.5 | 07.4/06.5/05.8 | 07.9/05.6/06.5 |

Table 4: **Standard Deviation Analysis** Standard deviations (STD) corresponding to Table 1. Each experiment shows three STD values separated by a backslash: (1) STD over 30 experiments with 10 random templates and 3 seeds, (2) mean STD over templates, and (3) mean STD over seeds.

# A  Standard Deviation

In table 4 we present the standard deviations (STD) corresponding to the main results shown in Table 1. For each experiment, we display three STD values, separated by a backslash. These values represent the variability in the results across different configurations:

1. The first value shows the standard deviation over 30 experiments, which includes 10 random templates and 3 seeds that determine the training examples. 2. The second value provides the mean of the standard deviation over the templates, the standard deviation across 10 templates, and the mean of the standard deviation across 3 seeds. 3. The third value presents the mean standard deviation over the seeds, the standard deviation over 3 seeds, and the mean over 10 templates.

This detailed breakdown of standard deviations allows for a more thorough understanding of the variability in model performance across different templates and seeds.

# B  Evaluation Details

All the graphs and ablation studies were conducted and evaluated using the DBPedia dataset with the GPT-J model. This setup was chosen due to the diversity of the DBPedia dataset, which includes a broad range of categories and entities, making it an ideal candidate for comprehensive evaluation. The use of GPT-J, a powerful generative model, ensures that the results are reflective of state-of-the-art performance in language modeling tasks. The combination of DBPedia and GPT-J allows us to thoroughly investigate the behavior of the model across various ablation settings, ensuring robust insights into the performance of different methods and configurations.

### B.1 Pruning for Classification

In our evaluation setup, we use pruning for classification by focusing only on the first token of the label, which is unique across all datasets. A common approach in the in-context learning setup is to iterate over all possible labels for each test sample and select the label with the highest probability according to the language model (LM). However, this approach can become computationally expensive, especially in cases where there are a large number of classes.

Similarly to Ratner et al. (2022), and given that the first token in each dataset is unique, we predict only the first token of the label and perform classification based on this value. While this approach deviates slightly from the common practice of iterating over all possible labels, the effect on the results should be minor.

### B.2 Test Set Size

For our experiments, we used a varying number of test examples depending on the dataset. Specifically, we used 100 test examples for the SST-2 dataset, and for datasets with a larger number of classes, the number of test examples was scaled linearly with the number of classes. For example, in the DBpedia dataset, which has 7 times more classes than SST-2, we used 700 test examples to ensure that the evaluation is proportional to the number of classes. This scaling helps to maintain a balanced evaluation across datasets with differing complexities, ensuring robust performance metrics for each method.

## C  Instruction Details

In some of the experiments, we use specific instructions to guide the model in performing the classification tasks. Below in table 5 that shows the instructions used for each dataset across all relevant methods:

| Dataset | Instruction |
|---------|-------------|
| SST2 | Classify the sentiment of the following text as positive or negative: |
| AG News | Classify the following text into one of the following categories: World, Sports, Business, Technology |
| DBpedia | Classify the following text into one of the following categories: Company, Educational Institution, Artist, Athlete, Office Holder, Mean Of Transportation, Building, Natural Place, Village, Animal, Plant, Album, Film, Written Work |
| TREC | Classify the following text into one of the following categories: Description, Entity, Expression, Human, Location, Number |

Table 5: Instructions used for relevant datasets in the experiments.

## D  Dataset Details

In our experiments, we used four different datasets, each representing a unique classification task. Table 6 provides an overview of the datasets and their respective tasks. Each dataset has a varying number of classes, denoted by $|C|$, which are detailed below:

- **SST-2**: This dataset is used for *sentiment analysis*, where the task is to classify movie reviews as either positive or negative. It contains 2 distinct classes.

- **AG News**: The AG News dataset is used for *news classification*. The task is to classify news articles into one of four categories: World, Sports, Business, and Technology. This dataset contains 4 classes.

- **DBpedia**: The DBpedia dataset is focused on *ontology classification*. The task involves classifying textual content into one of 14 distinct categories, which include entities such as Company, Artist, Village, and more.

- **TREC**: This dataset is used for *question classification*, where the goal is to classify questions into one of 6 answer types, including Description, Entity, Human, and Location.

Each dataset contains a specific number of examples based on its classification task, allowing us to evaluate the model's performance across a diverse range of challenges.

## E  Template Details

In our experiments, we use randomly selected templates from the options provided in table 7, suggested in Voronov et al. (2024). Each dataset is associated with both input and output templates, which are used to format the input data and the expected output during few-shot learning tasks.

- **Input Template**: As shown, this column lists the different templates for formatting the input data. For example, the SST-2 dataset uses "input: " and "text: " as input templates to introduce the input text.

- **Intra-Separator**: This separator is used between components (input and output) within a single example. For instance, AG News uses "\n" as an intra-separator between the input sentence and the output label.

- **Output Template**: The output template defines how the expected output is structured. For example, SST-2 employs formats like "output: , target: , label: " to guide the model in generating structured output.

- **Inter-Separator**: This column represents the separator used between multiple examples during training. In datasets like AG News and DBpedia, "\n\n" is used to separate examples.

We randomly select templates from the ones listed in table 7 for each experiment. This randomness in selecting templates introduces variability in the prompts, making the evaluation more robust and testing the model's ability to generalize across different input-output structures.

## F  Implementation Details

### F.1  Hyperparameter Details

In table 8 we present the hyperparameters used in our experiments across different models and datasets. The table provides the specific learning rates ('lr'), epsilon values ('$\epsilon$'), and format settings for the various methods applied to each dataset. The experiments were conducted using multiple model architectures, including **BLOOM 1.7B**, **GPT-J 6B**, and **LLaMA3 8B**, and we selected the best hyperparameters for each experiment: 2, 4, and 6 shots. Below is an overview of the key hyperparameters:

- **Learning Rate ('lr')**: The table provides the learning rates used for each method and dataset combination. For methods like *Prefix Tuning (PT)*, *Prompt Tuning (PT)*, *IPT*, and *LoRA*, learning rates vary from **1e-5** to **1e-3**, depending on the specific model and dataset.

| Dataset | Task | $|C|$ |
|---------|------|-------|
| SST-2 | Sentiment analysis (movie) | 2 |
| AG News | News classification (topic) | 4 |
| DBpedia | Ontology classification | 14 |
| TREC | Question classification (answer type) | 6 |

Table 6:  **Dataset Overview** These are the datasets used, representing a range of different types of classification tasks, including SST-2, AG News, DBpedia, and TREC. Each dataset has a varying number of classes (denoted by $|C|$).

| Dataset | Input Template | Intra-Separator | Output Template | Inter-Separator |
|---------|----------------|-----------------|-----------------|-----------------|
| SST-2 | "input: {}", "text: ", | " ", "\n" | "output: {}", "target: {}", "label: {}", "emotion: {}", "semtiment: {}", "A {} one.", "It was {}.", "All in all {}.", "A {} piece." | " ", ",", "\n", "\n\n" |
| AG News | "sentence: {}", "{}" | | "output: {}", "target: {}", "label: {}", "Topic: {}.", "Subject: {}.", "This is about {}.", "It is about {}." | |
| DBpedia | | | | |
| TREC | | | | |

Table 7: **Template Options for Various Datasets** We provide various template options for different datasets. Each dataset include both input and output templates, and also includes intra-separators between inputs and labels, as well as inter-separators between examples.

- **CPT Hyperparameters**: For *CPT*, we also report epsilon values ('$\epsilon$') for both the *input* and the *format* components. These epsilon values control the magnitude of the perturbations applied during optimization. The values of epsilon vary across different models and datasets, generally ranging from **1e-2** to **1e-0** for both input and format components.

- **Model Variability**: The table reflects variability in hyperparameter choices depending on the model size and architecture. For instance, *GPT-3 6B* typically requires higher learning rates compared to *BLOOM 1.7B*, as seen with *CPT* and other methods. The hyperparameters are carefully tuned to optimize performance on tasks such as SST-2, AG News, DBpedia, and TREC.

These hyperparameters are critical for achieving optimal performance in few-shot learning settings. They control the learning process, model updates, and how much the model is allowed to adapt to new data. The values in table 8 are based on extensive experimentation and fine-tuning to ensure the best results for each method and dataset.

### F.2 Methods Implementation Details

In our experiments, we utilized existing implementations for several methods and implemented IPT ourselves. Specifically, we used the implementations provided by the *Parameter-Efficient Fine-Tuning Mangrulkar et al. (2022) (PEFT)* library [1] for methods such as **LoRA**, **Prefix Tuning**, and **Prompt Tuning (PT)**. For IPT, we built our implementation based on the PEFT framework.

For all experiments, we used the recommended parameters:

- For LoRA, we set $\alpha = 16$ and the rank $r = 8$.

- For Prompt Tuning, Prefix Tuning, and IPT we used 8 learnable tokens.

By using the PEFT framework, we ensure that our fine-tuning processes for LoRA, Prefix Tuning, and PT are aligned with current standards, while our custom IPT implementation extends the framework to allow for additional flexibility in parameter-efficient training.

### F.3 Training Details

We utilized the 'Fine-tune a pretrained model' package from Wolf et al. (2020), which provides a comprehensive framework for training and evaluating models[2]. For all baselines, we employed the default parameters provided by the trainer, ensuring consistency across experiments. Each model was trained for 25 epochs, allowing sufficient time for convergence while maintaining uniform training conditions across methods.

---

[1] https://huggingface.co/docs/peft/en/index
[2] https://huggingface.co/docs/transformers/en/training

| Dataset | Method | Paremeter | BLOOM 1.7B | | | GPT-J 6B | | | LLaMA3 8B | | |
|---|---|---|---|---|---|---|---|---|---|---|---|
| | | | 2 | 4 | 6 | 2 | 4 | 6 | 2 | 4 | 6 |
| SST-2 | Prefix Tuning | lr | $1e-3$ | $1e-3$ | $1e-3$ | $1e-5$ | $1e-4$ | $1e-3$ | $-$ | $-$ | $-$ |
| | PT† | lr | $1e-5$ | $1e-5$ | $1e-5$ | $1e-4$ | $1e-3$ | $1e-3$ | $1e-5$ | $1e-5$ | $1e-5$ |
| | PT | lr | $1e-5$ | $1e-5$ | $1e-5$ | $1e-5$ | $1e-5$ | $1e-5$ | $1e-5$ | $1e-5$ | $1e-5$ |
| | IPT† | lr | $1e-5$ | $1e-4$ | $1e-4$ | $1e-5$ | $1e-3$ | $1e-4$ | $1e-5$ | $1e-5$ | $1e-4$ |
| | IPT | lr | $1e-5$ | $1e-5$ | $1e-5$ | $1e-5$ | $1e-4$ | $1e-4$ | $1e-5$ | $1e-5$ | $1e-5$ |
| | LoRA | lr | $1e-5$ | $1e-5$ | $1e-5$ | $1e-5$ | $1e-5$ | $1e-5$ | $1e-5$ | $1e-4$ | $1e-4$ |
| | CPT† | lr | $1e-5$ | $1e-3$ | $1e-4$ | $1e-5$ | $1e-4$ | $1e-3$ | $1e-5$ | $1e-5$ | $1e-5$ |
| | | Input $\epsilon$ | $1e-3$ | $1e-0$ | $1e-0$ | $1e-3$ | $1e-1$ | $1e-1$ | $1e-1$ | $1e-1$ | $1e-0$ |
| | | Format $\epsilon$ | $1e-3$ | $1e-3$ | $1e-3$ | $1e-3$ | $1e-2$ | $1e-3$ | $1e-2$ | $1e-1$ | $1e-0$ |
| | CPT | lr | $1e-3$ | $1e-3$ | $1e-4$ | $1e-5$ | $1e-4$ | $1e-4$ | $1e-3$ | $1e-4$ | $1e-4$ |
| | | Input $\epsilon$ | $1e-2$ | $1e-0$ | $1e-0$ | $1e-3$ | $1e-0$ | $1e-0$ | $1e-2$ | $1e-0$ | $1e-2$ |
| | | Format $\epsilon$ | $1e-2$ | $1e-2$ | $1e-3$ | $1e-3$ | $1e-3$ | $1e-2$ | $1e-3$ | $1e-3$ | $1e-3$ |
| AG News | Prefix Tuning | lr | $1e-4$ | $1e-3$ | $1e-3$ | $1e-5$ | $1e-5$ | $1e-3$ | $-$ | $-$ | $-$ |
| | PT† | lr | $1e-3$ | $1e-3$ | $1e-3$ | $1e-5$ | $1e-3$ | $1e-3$ | $1e-4$ | $1e-4$ | $1e-4$ |
| | PT | lr | $1e-3$ | $1e-3$ | $1e-3$ | $1e-4$ | $1e-3$ | $1e-3$ | $1e-4$ | $1e-5$ | $1e-4$ |
| | IPT† | lr | $1e-3$ | $1e-3$ | $1e-3$ | $1e-5$ | $1e-4$ | $1e-4$ | $1e-4$ | $1e-5$ | $1e-5$ |
| | IPT | lr | $1e-4$ | $1e-3$ | $1e-4$ | $1e-5$ | $1e-5$ | $1e-4$ | $1e-5$ | $1e-5$ | $1e-5$ |
| | LoRA | lr | $1e-5$ | $1e-4$ | $1e-3$ | $1e-5$ | $1e-5$ | $1e-4$ | $1e-5$ | $1e-5$ | $1e-5$ |
| | CPT† | lr | $1e-4$ | $1e-3$ | $1e-3$ | $1e-4$ | $1e-4$ | $1e-4$ | $1e-5$ | $1e-5$ | $1e-5$ |
| | | Input $\epsilon$ | $1e-2$ | $1e-0$ | $1e-0$ | $1e-1$ | $1e-1$ | $1e-2$ | $1e-1$ | $1e-3$ | $1e-3$ |
| | | Format $\epsilon$ | $1e-1$ | $1e-2$ | $1e-0$ | $1e-1$ | $1e-3$ | $1e-0$ | $1e-1$ | $1e-2$ | $1e-3$ |
| | CPT | lr | $1e-4$ | $1e-4$ | $1e-3$ | $1e-3$ | $1e-4$ | $1e-4$ | $1e-3$ | $1e-4$ | $1e-3$ |
| | | Input $\epsilon$ | $1e-2$ | $1e-0$ | $1e-0$ | $1e-2$ | $1e-0$ | $1e-0$ | $1e-2$ | $1e-3$ | $1e-3$ |
| | | Format $\epsilon$ | $1e-2$ | $1e-0$ | $1e-0$ | $1e-3$ | $1e-3$ | $1e-0$ | $1e-3$ | $1e-3$ | $1e-3$ |
| DBpedia | Prefix Tuning | lr | $1e-3$ | $1e-3$ | $1e-3$ | $1e-3$ | $1e-3$ | $1e-3$ | $-$ | $-$ | $-$ |
| | PT† | lr | $1e-3$ | $1e-5$ | $1e-3$ | $1e-5$ | $1e-3$ | $1e-3$ | $1e-4$ | $1e-4$ | $1e-4$ |
| | PT | lr | $1e-4$ | $1e-5$ | $1e-4$ | $1e-3$ | $1e-3$ | $1e-3$ | $1e-4$ | $1e-5$ | $1e-5$ |
| | IPT† | lr | $1e-4$ | $1e-5$ | $1e-5$ | $1e-5$ | $1e-4$ | $1e-5$ | $1e-5$ | $1e-5$ | $1e-5$ |
| | IPT | lr | $1e-5$ | $1e-5$ | $1e-5$ | $1e-5$ | $1e-5$ | $1e-5$ | $1e-5$ | $1e-5$ | $1e-5$ |
| | LoRA | lr | $1e-4$ | $1e-5$ | $1e-5$ | $1e-4$ | $1e-4$ | $1e-4$ | $1e-5$ | $1e-5$ | $1e-5$ |
| | CPT† | lr | $1e-5$ | $1e-5$ | $1e-5$ | $1e-4$ | $1e-5$ | $1e-5$ | $1e-5$ | $1e-5$ | $1e-5$ |
| | | Input $\epsilon$ | $1e-2$ | $1e-2$ | $1e-1$ | $1e-0$ | $1e-1$ | $1e-1$ | $1e-0$ | $1e-1$ | $1e-1$ |
| | | Format $\epsilon$ | $1e-1$ | $1e-0$ | $1e-1$ | $1e-3$ | $1e-0$ | $1e-1$ | $1e-1$ | $1e-0$ | $1e-1$ |
| | CPT | lr | $1e-4$ | $1e-4$ | $1e-5$ | $1e-4$ | $1e-4$ | $1e-4$ | $1e-5$ | $1e-5$ | $1e-5$ |
| | | Input $\epsilon$ | $1e-0$ | $1e-2$ | $1e-0$ | $1e-0$ | $1e-0$ | $1e-0$ | $1e-2$ | $1e-0$ | $1e-3$ |
| | | Format $\epsilon$ | $1e-0$ | $1e-0$ | $1e-0$ | $1e-0$ | $1e-3$ | $1e-3$ | $1e-2$ | $1e-3$ | $1e-2$ |
| TREC | Prefix Tuning | lr | $1e-3$ | $1e-3$ | $1e-3$ | $1e-3$ | $1e-3$ | $1e-5$ | $-$ | $-$ | $-$ |
| | PT† | lr | $1e-3$ | $1e-3$ | $1e-3$ | $1e-3$ | $1e-3$ | $1e-3$ | $1e-4$ | $1e-4$ | $1e-4$ |
| | PT | lr | $1e-5$ | $1e-3$ | $1e-3$ | $1e-5$ | $1e-3$ | $1e-3$ | $1e-5$ | $1e-5$ | $1e-5$ |
| | IPT† | lr | $1e-3$ | $1e-3$ | $1e-3$ | $1e-4$ | $1e-3$ | $1e-4$ | $1e-4$ | $1e-4$ | $1e-5$ |
| | IPT | lr | $1e-5$ | $1e-3$ | $1e-3$ | $1e-4$ | $1e-4$ | $1e-4$ | $1e-5$ | $1e-5$ | $1e-5$ |
| | LoRA | lr | $1e-4$ | $1e-5$ | $1e-5$ | $1e-5$ | $1e-5$ | $1e-4$ | $1e-5$ | $1e-5$ | $1e-4$ |
| | CPT† | lr | $1e-3$ | $1e-3$ | $1e-3$ | $1e-4$ | $1e-4$ | $1e-x$ | $1e-4$ | $1e-5$ | $1e-5$ |
| | | Input $\epsilon$ | $1e-0$ | $1e-0$ | $1e-0$ | $1e-1$ | $1e-0$ | $1e-0$ | $1e-1$ | $1e-1$ | $1e-1$ |
| | | Format $\epsilon$ | $1e-3$ | $1e-1$ | $1e-2$ | $1e-1$ | $1e-0$ | $1e-2$ | $1e-3$ | $1e-0$ | $1e-0$ |
| | CPT | lr | $1e-3$ | $1e-3$ | $1e-4$ | $1e-3$ | $1e-3$ | $1e-3$ | $1e-4$ | $1e-4$ | $1e-4$ |
| | | Input $\epsilon$ | $1e-0$ | $1e-0$ | $1e-0$ | $1e-0$ | $1e-0$ | $1e-3$ | $1e-0$ | $1e-0$ | $1e-0$ |
| | | Format $\epsilon$ | $1e-0$ | $1e-0$ | $1e-3$ | $1e-2$ | $1e-2$ | $1e-0$ | $1e-2$ | $1e-3$ | $1e-0$ |

Table 8: **Hyperparameters** Hyperparameters used for each experiment across 2, 4, and 6 shots for different models, including BLOOM 1.7B, GPT-J 6B, and LLaMA3 8B. The table shows learning rates (lr), epsilon values for input and format, and other parameters for methods such as Prefix Tuning, Prompt Tuning, IPT, LoRA, and CPT. The experiments were conducted on datasets like SST-2, AG News, DBpedia, and TREC.

| Method | Prefix Construction |
|---|---|
| **PT** | In this part, we use only random embedding initialization. |
| **PT**† | Classify the sentiment of the following text as positive or negative. |
| **IPT** | In this part, we use only random embedding initialization. input: the greatest musicians output: positive. input: the action is stilted output: negative. |
| **IPT**† | Classify the sentiment of the following text as positive or negative. input: the greatest musicians output: positive. input: the action is stilted output: negative. |
| **CPT** | input: the greatest musicians output: positive. input: the action is stilted output: negative. |
| **CPT**† | Classify the sentiment of the following text as positive or negative. input: the greatest musicians output: positive. input: the action is stilted output: negative. |

Table 9: Input Construction for PT, IPT, and CPT (with and without †) using SST-2. The updated text during training is marked in red.

## G  Input Preparation

In this section, we provide a detailed explanation of how the input is constructed for different methods, including Prompt Tuning (PT), Instruction Prompt Tuning (IPT), and Context-Aware Prompt Tuning (CPT), both with and without the † variant. To clarify the differences, we use SST-2 as an example with the instruction: *"Classify the sentiment of the following text as positive or negative."*

Each example is constructed using a template that includes `input:` and `output:`, where the `input` corresponds to the actual text of the example, and the `output` corresponds to its label. For instance:

- **Example 1:** The `input` is "`the greatest musicians`", and the `output` is "`positive`".

- **Example 2:** The `input` is "`the action is stilted`", and the `output` is "`negative`".

Using the template, these examples are represented as:

- **Example 1:** `input:  the greatest musicians output:  positive`

- **Example 2:** `input:  the action is stilted output:  negative`

This template-based construction ensures consistency across the methods, allowing us to clearly define how the input and output are represented in different approaches, such as PT, IPT, and CPT.

table 9 outlines the construction of the prefix for each method and highlights which parts are updated during training.

## H  Projected Gradient Descent (PGD) Algorithm

In our method, we initialize the context tokens, denoted as $x_i$, using the training examples, with each token $x_i$ associated with a vector $\delta_i$, which is initially set to zero. For simplicity, we use $x_i$ and $\delta_i$ to denote these components only in this part of the explanation.

During the optimization process, the tokens $x_i$ remain fixed, while the $\delta_i$ vectors are updated iteratively. After each optimizer update, we perform a post-processing step where each $\delta_i$ is projected to ensure that its L2 norm does not exceed a predefined limit, $\epsilon$. It is important to note that this projection step is independent of the optimizer and serves as an additional operation to control the extent of change for each context token.

1: Initialize each $\delta_i \leftarrow 0$
2: Initialize $x_i \leftarrow$ `training_examples_tokens`
3: **for** $j \leftarrow 1$ to `num_of_training_steps` **do**
4:     $\delta_i \leftarrow \delta_i - \alpha \nabla \text{Loss}(f(x_i + \delta_i), y_i)$                                      $\triangleright$ Gradient descent step
5:     $n_i \leftarrow \|\delta_i\|$                                           $\triangleright$ Compute the L2 norm of $\delta_i$
6:     $\delta_i \leftarrow \delta_i \times \text{clip}(n_i, \epsilon)/n_i$                        $\triangleright$ Project $\delta_i$ to ensure L2 norm $\leq \epsilon$
7: **end for**

This ensures that the updates to $\delta_i$ remain constrained, preventing excessive modifications to the context tokens and maintaining a balance between optimization and regularization. The process allows the model to adapt while ensuring that changes to the context tokens remain meaningful and controlled.

## I   Evaluating the Impact of Projected Gradient Descent (PGD)

Our method use the same optimizer used for all baselines. However, our method incorporates an additional step after each parameter update: we project each token, restricting its allowed change. The allowed change is determined by the hyperparameters `Input` $\epsilon$ and `Format` $\epsilon$, which define the L2 norm limit for each token's modification.

To ensure that PGD Madry et al. (2017) is not the sole reason for our method's improvement, we conducted two types of experiments. First, we compared our method without PGD to PT and IPT. Second, we added a PGD step to PT and IPT for comparison.

For the first experiment, we compared CPT (without PGD) to PT and IPT on the DBpedia dataset. The results for 2, 4, and 6 shots are presented in Table 10.

| Method | 2 Shots | 4 Shots | 6 Shots |
|---|---|---|---|
| PT | 23.39 | 29.69 | 40.53 |
| IPT | 52.86 | 67.27 | 70.73 |
| CPT (No PGD) | 68.28 | 74.17 | 77.52 |

Table 10: Performance Comparison Without PGD (DBpedia), using GPT-J.

For the second experiment, we compared CPT† to PT† and IPT† (with and without PGD) on the DBpedia dataset. To ensure a fair comparison, we performed hyperparameter tuning (HPT) over $\epsilon$ and the learning rate for both PT and IPT. The results for 2, 4, and 6 shots are presented in Table 11.

| Method | 2 Shots | 4 Shots | 6 Shots |
|---|---|---|---|
| PT† | 12.96 | 22.12 | 37.44 |
| PT† + PGD | 12.80 | 22.02 | 38.69 |
| IPT† | 47.10 | 66.37 | 75.09 |
| IPT† + PGD | 47.10 | 66.40 | 75.09 |
| CPT† + PGD | 52.87 | 77.30 | 81.00 |

Table 11: Performance Comparison With and Without PGD (DBpedia), using GPT-J.

The results clearly demonstrate that, in both experiments, our method consistently outperforms PT and IPT. Furthermore, it is evident that other methods do not necessarily benefit from the addition of PGD. While we cannot definitively explain this, we hypothesize that it may be due to the highly effective way in which we employ PGD, leveraging prior knowledge about the structure of the input, format, and labels within the context. Our approach allows us to apply distinct projections to different components of the context, which we believe significantly contributes to the superior performance of our method.

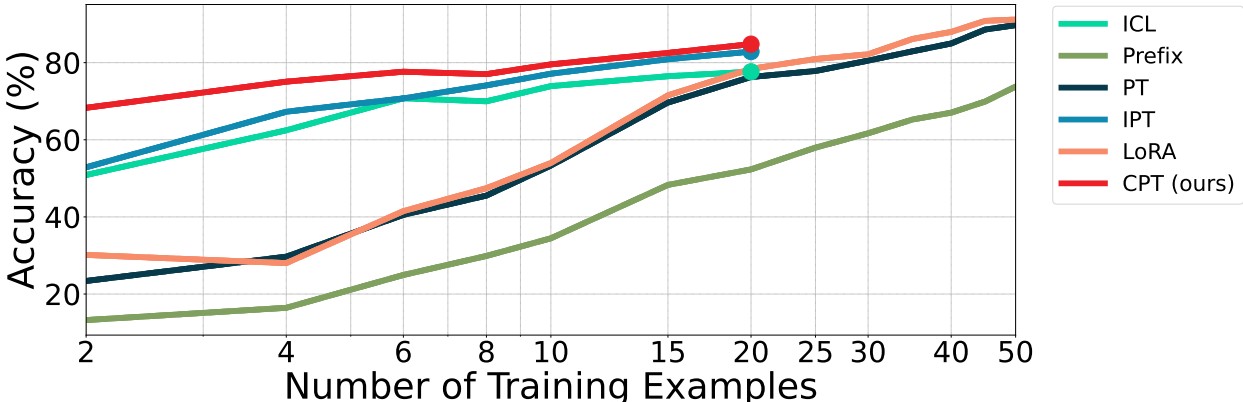

Figure 6: **Comparison of Few-Shot Methods.** We compare CPT with baseline methods using the GPT-J model and the DBpedia dataset in few-shot settings, demonstrating its superior performance, particularly when handling a limited number of examples. Furthermore, our results highlight that context-based methods encounter memory constraints (indicated by dots) as the number of training examples increases beyond a certain threshold.

## J  Large Number of Training Examples

## K  Latency and Memory Consumption

As shown in fig. 7, the inference latency and memory consumption of the four methods—LoRA, CPT (ours), IPT, and PT—exhibit distinct patterns as the number of training examples increases. In terms of latency, LoRA consistently maintains the lowest and most stable inference time, which can be attributed to its fixed input sequence length. PT also has a fixed input length and therefore does not experience latency increases due to sequence growth, though minor fluctuations may occur from runtime variability. In contrast, CPT

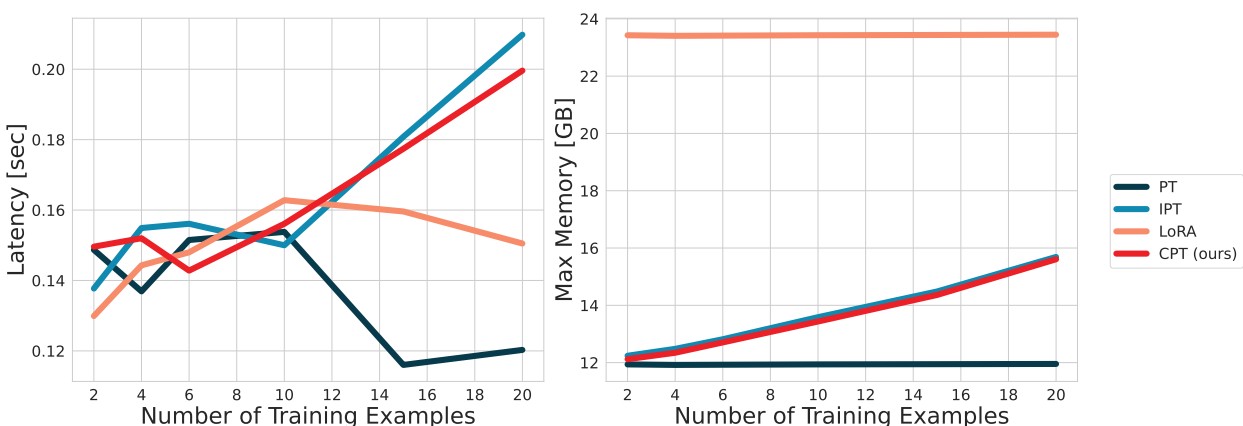

Figure 7: **Latency and Memory Usage Analysis.** Left: Inference latency (seconds) vs. number of training examples. LoRA exhibits the lowest and most stable latency, while CPT (ours), IPT, and PT show increasing or fluctuating trends. Right: Maximum memory usage (GB) during training. LoRA's memory remains constant but significantly higher than the other methods; PT's memory is also constant, while CPT (ours) and IPT increase linearly with the number of training examples.

| Dataset | BLOOM 1.7B | | | GPT-J 6B | | | LLaMA3 8B | | |
|---------|-----|-----|-----|-----|-----|-----|-----|-----|-----|
| | 2 | 4 | 6 | 2 | 4 | 6 | 2 | 4 | 6 |
| SST-2 | 67.40 | 67.63 | 67.73 | 71.40 | 71.83 | 71.96 | 74.53 | 76.06 | 80.13 |
| AG News | 36.01 | 39.16 | 52.15 | 57.51 | 57.23 | 68.25 | 75.48 | 72.00 | 74.00 |
| DBpedia | 46.24 | 48.16 | 51.79 | 34.77 | 37.94 | 48.66 | 62.10 | 64.32 | 66.01 |
| TREC | 33.30 | 33.97 | 35.33 | 34.35 | 34.37 | 34.34 | 33.65 | 35.04 | 34.86 |

Table 12: **IA3³** parameter-efficient fine-tuning method across varying numbers of shots (2, 4, 6) on four classification datasets using BLOOM 1.7B, GPT-J 6B, and LLaMA3 8B models.

and IPT show a clear upward latency trend as the number of training examples grows, driven by their increasing input sequence lengths, which require more computation during the forward pass.

Regarding memory usage during training, LoRA stands out by requiring a significantly higher but constant amount of memory, as it optimizes low-rank adaptation matrices while keeping the base model weights frozen. PT also shows constant memory consumption because it only fine-tunes a fixed set of prompt embeddings without altering the underlying model parameters, thus maintaining a low and stable memory footprint regardless of dataset size. On the other hand, CPT and IPT experience a roughly linear increase in memory usage as training examples grow, reflecting the additional storage needed for their more complex prompt or context embedding mechanisms. This comparison highlights LoRA's stability in latency and memory, PT's minimal memory footprint, and the trade-offs CPT and IPT make between adaptability and resource consumption.

## L  Evaluation of IA$^3$

As shown in table 12 Liu et al. (2022), across all evaluated models (BLOOM 1.7B, GPT-J 6B, and LLaMA3 8B) and datasets (SST-2, AG News, DBpedia, and TREC), IA$^3$ consistently underperforms compared to our proposed method. While IA$^3$ benefits from a lightweight parameter-efficient adaptation, its limited flexibility in modifying the underlying model's representations appears to hinder performance, particularly in low-shot settings. In contrast, our approach achieves superior accuracy in every configuration, highlighting its stronger ability to leverage few-shot examples and adapt context effectively across diverse architectures and tasks.

