# OpenReview forum: "Context-aware Prompt Tuning: Enhancing Few-Shot Learning via Optimized Context Embeddings"
_TMLR — Rejected by TMLR_

### Review · Reviewer_UxCJ · 2025-07-25

**Summary Of Contributions:**

This paper addresses a critical gap in few-shot learning: the underperformance of optimization-based methods (e.g., Prompt Tuning, LoRA) in low-data regimes due to overfitting, while In-Context Learning (ICL) excels but plateaus with more data. The authors propose Context-Aware Prompt Tuning (CPT), a novel approach that combines ICL’s robustness with a constrained optimization strategy to mitigate overfitting.  The authors conduct extensive experiments on various models (e.g.,BLOOM 1.7B, GPT-J 6B, LLaMA3 8B)

**Audience:**

Yes

**Audience Explanation:**

1 CPT bridges ICL and optimization-based methods, offering a principled way to balance flexibility and regularization—an active area of research
2 The approach improves performance in low-data scenarios, which is critical for real-world applications where labeled data is scarce.

**Claims And Evidence:**

Yes

**Claims Explanation:**

The authors provide extensive experiments across models (BLOOM, GPT-J, LLaMA3) and tasks, showing CPT’s consistent outperformance . Ablation studies effectively isolate the impact of key components (e.g., loss design, PGD projections).

**Requested Changes:**

-Would the method work well on a Moe models? As moe with both efficiency and high capacity gradually becomes the mainstream, it is better to conduct experiments on these models.

-Expand on computational overhead (Discussion) by quantifying training time/memory usage relative to baselines. Will the proposed method increase the computational cost？

---

> ### Author Response · Authors · 2025-08-09
> **Answering UxCJ**
>
> We thank the reviewer for their thoughtful and encouraging feedback, and for highlighting CPT’s strength in “balancing flexibility and regularization” to improve performance in low-data scenarios. As noted, CPT is a novel approach that combines the robustness of context-based learning with a constrained optimization strategy, mitigating overfitting while preserving interpretability.
>
> Regarding the request to quantify training latency and memory usage relative to baselines, we have added a detailed analysis in the Appendix K and discussed it in the Discussion section. The results compare CPT against Prompt Tuning (PT), Instruction Prompt Tuning (IPT), and LoRA. We find that CPT’s memory usage grows linearly with the number of training examples, similar to IPT, while latency is comparable to IPT due to the growing input size. LoRA maintains a stable but significantly higher memory footprint than all prompt-tuning methods. This expanded evaluation directly addresses the reviewer’s concern by quantifying CPT’s computational overhead and showing that its performance gains come with manageable resource requirements.
>
> Regarding the question about MoE models, we agree that this is indeed an interesting and promising direction. However, it is outside the scope of the current work, as our study already covers a broad and diverse evaluation setup involving three different model families, two types of tasks (classification and question answering), and seven datasets. This comprehensive evaluation was necessary to rigorously validate CPT’s effectiveness and robustness across architectures, task types, and domains, but it also required substantial experimental resources. A thorough investigation of CPT’s applicability to MoE architectures would require a similarly extensive experimental design, which we leave for future work.

---

### Review · Reviewer_6mED · 2025-08-01

**Summary Of Contributions:**

This paper focuses on enhancing the performance of few-shot learning of LLM with in-context learner by incorporating optimization into ICL. The authors first perform experiments and show that the overfitting problem is severer in optimization few-shot learning methods than in in-context learning while the optimization approaches performs well when there are more samples. To this end, the authors propose Context-Aware Prompt Tuning (CPT) that extends ICL with the designed optimization w.r.t. the contexts. They use loss over training samples and context samples and introduce a loss weighting scheme, only updating contexts tokens, and projected gradient descent for preventing overfitting. Results on several datasets show that the proposed method obtains better results in most cases while it is worse in several cases.

**Audience:**

Yes

**Audience Explanation:**

The paper focuses on enhancing the performance of ICL with optimization strategies in very low sample regime and there should be individuals in TMLR's audience interested in the findings of the paper.

**Broader Impact Concerns:**

The paper introduces new method for improving few-shot learning results of LLM. Misusing the technique can lead to ethical implications. I would recommend the authors to include a section of Broader Impact Concerns

**Claims And Evidence:**

Yes

**Claims Explanation:**

The authors claim that overfitting is the issue of existing optimization few-shot learning methods for obtaining good performance in very low sample regime and this is supported by the results. The results also show that their method obtains better results in most cases. More details can be found in Requested changes.

**Requested Changes:**

## strengths & Weaknesses

+ ***Strengths***

1. The problem of enhancing the ability of methods in extreme low sample regime is interesting and can be helpful for practical applications.

2. It is interesting to identify the overfitting problem in few-shot learning.

3. The method of integrating ICL and optimization is interesting.

- ***Weaknesses***

1. The paper is hard to follow and I would suggest the authors to better organise and present the paper. For example, I would suggest to clearly explain the few-shot learning settings, ICL and optimization first and then explain their approach. Additionally, the proposed method is not clearly presented. For example, it is not clear how the projected gradient descent performed for changing the gradient magnitude. What kind of conflicts when using contexts for two roles? What parameters/tokens are optimised for minimising the training loss? - The training loss are computed over training tokens and the training tokens and the model are frozen.

2. The authors claim that optimization is worse than ICL in few-shot settings while Liu et al., 2022 has shown that few-shot PEFT methods is better than ICL in few-shot learning. Any discussions and comparisons between (Liu et al., 2022) and the proposed method?

3. In figure 1, the gap between training and testing loss increases when the number of samples increases for CPT. Why the gap is smaller in less samples for CPT?

4. From the ablation study, it seems that the projected gradient descent component has the largest impact on the results instead of having ICL and it seems that projected gradient descent is not limited to ICL and may help preventing optimization methods.



Liu, Haokun, et al. "Few-shot parameter-efficient fine-tuning is better and cheaper than in-context learning." Advances in Neural Information Processing Systems 35 (2022): 1950-1965.

## Requested Changes

1. Reorganise and better present the paper to more clearly explain the methods and the idea.

2. Adding discussions and comparisons to few-shot PEFT methods such as (Liu et al., 2022).

3. Solving some unclear parts in the paper as mentioned in the weakness section.

3. Proofread the paper.

---

> ### Author Response · Authors · 2025-08-10
> **Answering 6mED**
>
> We thank the reviewer for their comments and suggestions, which helped us strengthen the paper.
>
> Reminder of the paper’s focus and contributions
> Our work proposes Context-aware Prompt Tuning (CPT), a method that bridges the strengths of In-Context Learning (ICL) and parameter-efficient optimization. While pure ICL often outperforms standard optimization-based approaches in very low-shot regimes, its effectiveness declines as the number of shots grows, and methods like IA³ are typically evaluated with ≥20 training examples. In contrast, our focus is on extremely low-shot scenarios (2–6 examples), where ICL is naturally strong.
>
> CPT enhances ICL by introducing a specialized optimization procedure that iteratively refines the context embeddings while keeping the underlying language model completely frozen. Only the embeddings corresponding to the in-context examples are updated. This design preserves the interpretability of ICL — ensured by a projection step that limits embedding changes — while improving adaptability to the specific task. The construction of the input and the optimization process are illustrated in Figures 3 and 4.
>
> In response to the reviewer’s suggestion, we have now also included IA³ as an additional baseline (Appendix L). Across all evaluated models (BLOOM 1.7B, GPT-J 6B, and LLaMA3 8B) and datasets (SST-2, AG News, DBpedia, and TREC), IA³ consistently underperforms CPT in these very low-shot settings. This supports our central claim that enhancing ICL with our targeted optimization procedure is the preferred approach in such regimes.
>
> We believe these clarifications and the added IA³ results directly address the reviewer’s concern while confirming our original conclusions, without requiring changes to the main text of the paper.

---

### Review · Reviewer_tSk9 · 2025-08-04

**Summary Of Contributions:**

**Summary Of contributions**:
- Diagnosed overfitting as the main bottleneck of existing optimization-based few-shot methods.
- Proposed CPT, which integrates ICL with a parameter-efficient, regularized optimization process.
- Demonstrated SOTA performance across multiple tasks and models, and provided comprehensive ablation studies.

**Strengths**:
- The paper proposes CPT, a novel hybrid method that integrates the strengths of ICL and prompt optimization with a tailored regularization strategy to mitigate overfitting.
- CPT is evaluated across multiple datasets (classification + QA), three model families, and several few-shot settings (2, 4, 6 shots).

**Weaknesses**:
- Sections 3.2–3.3 describing input construction and optimization steps are dense and formula-heavy, which may hinder comprehension. Clearer diagrams or more intuitive summaries would improve accessibility.
- A major motivation for CPT is maintaining the interpretability of context tokens. However, there is no qualitative analysis or visualization (e.g., token embedding drift, example outputs) to show that this interpretability is preserved.
- The paper does not explore how CPT performs on or extends to instruction-tuned models, which dominate many NLP applications today.

**Audience:**

Yes

**Audience Explanation:**

Yes, at least some individuals in TMLR’s audience would be highly interested in the findings of this paper. The paper addresses a central and timely challenge in the field of LLMs: improving performance in few-shot learning scenarios. Moreover, the proposed method—CPT—offers a novel, practical alternative to conventional prompt tuning and fine-tuning strategies by effectively combining the generalization strengths of ICL with parameter-efficient optimization. These contributions are especially relevant as the field increasingly explores low-resource and scalable adaptation methods for large models.

**Claims And Evidence:**

Yes

**Claims Explanation:**

The claims in the paper are well-supported by accurate and convincing evidence. The authors conduct extensive experiments across multiple datasets (e.g., SST-2, DBpedia, BoolQ), models (BLOOM, GPT-J, LLaMA3), and few-shot settings (2, 4, 6 shots), consistently demonstrating that their proposed method, CPT, outperforms strong baselines. They rigorously account for variability using 30-run averages and provide detailed ablation studies showing the impact of each component (e.g., loss design, projection type). The methodology is clearly explained with theoretical justification and visual aids. Additionally, the paper transparently acknowledges limitations like computational cost and prompt length constraints. Overall, the evidence is clear, robust, and supports the authors' claims.

**Requested Changes:**

- The method requires iterative optimization, which incurs additional cost compared to ICL, but this is only briefly acknowledged. They are recommended to provide a more detailed analysis or quantification of CPT’s computational cost (e.g., runtime per training instance, memory usage), ideally compared against LoRA, PT, and ICL.
- The method is framed strictly in the few-shot context, but its design (context-based embeddings + PGD) may generalize. They should briefly discuss whether CPT could extend to medium-shot or continual learning settings, even if not empirically tested.
- The exponential decay weighting based on context position is adopted from prior work but not deeply justified. The intuitive reasoning or a toy example showing how position affects model behavior and why recency bias helps.

---

> ### Author Response · Authors · 2025-08-09
> **Answering tSk9**
>
> We thank the reviewer for their constructive and detailed feedback. As the reviewer noted, CPT is “a novel hybrid method that integrates the strengths of ICL and prompt optimization with a tailored regularization strategy to mitigate overfitting”, and the paper “demonstrated SOTA performance across multiple tasks and models, and provided comprehensive ablation studies”.
>
> (1) “Provide a more detailed analysis or quantification of CPT’s computational cost (e.g., runtime per training instance, memory usage), ideally compared against LoRA, PT, and ICL.”
> We have added a detailed computational analysis in the Appendix K and discussed it in the main text (Discussion). The new figure reports training latency and peak memory usage for CPT compared to PT, IPT, and LoRA. CPT’s latency trend is similar to prompt-based baselines and remains close to IPT, while LoRA exhibits a higher but stable memory footprint. A key limitation—shared with prompt-based methods—is that more shots increase the input sequence length, and standard attention causes quadratic memory growth with sequence length, making very high-shot settings less efficient.
>
> (2) “Should briefly discuss whether CPT could extend to medium-shot or continual learning settings, even if not empirically tested.”
> While CPT significantly improves performance, it incurs additional computational cost due to the iterative optimization of context embeddings, and is therefore less suited for larger training sets or longer input sequences. We also refer the reader to the Limitations section, where we discuss the challenges of applying CPTto medium-shot or continual learning scenarios and the need for more efficient handling of longer contexts.
>
> (3) “The exponential decay weighting based on context position is adopted from prior work but not deeply justified.”
> The recency bias effect underlying exponential decay weighting has been empirically demonstrated in prior work, showing that examples closer to the prediction position have greater influence on generation. We leverage this established finding and show in our ablations that incorporating exponential decay weighting into CPT’s loss function further enhances optimization performance.

---

### Author Response · Authors · 2025-08-10
**Response to all reviewers**

We thank all reviewers for their constructive feedback, which helped us strengthen the paper.

This work proposes Context-Aware Prompt Tuning (CPT), a method that combines the strengths of In-Context Learning (ICL) and parameter-efficient optimization. In extremely low-shot regimes (2–6 examples), ICL is known to outperform standard optimization-based methods (and we also show it), while approaches like IA³ are typically evaluated with 20 or more examples. CPT enhances ICL by iteratively optimizing only the embeddings of the in-context examples, keeping the base model frozen, and applying a projection step to preserve interpretability. Figures 3 and 4 illustrate the input construction and optimization process.

In response to reviewer suggestions, we added a detailed computational analysis (Appendix K) comparing CPT to PT, IPT, and LoRA in terms of latency and memory usage, included IA³ results (Appendix L) across BLOOM 1.7B, GPT-J 6B, and LLaMA3 8B showing that IA³ consistently underperforms CPT in these settings. We expanded the limitations discussion to address possible extensions to medium-shot and continual learning, and clarified the design choices, including the motivation for exponential decay weighting. These additions address all specific reviewer requests while preserving the paper’s core structure.

---

### Decision · Action_Editor_kGGG · 2025-09-05

**Recommendation:** Reject

**Additional Comments:**

After going through the paper and reviewers comments, I believe that this work has potential. Nevertheless, in its current shape it still requires some work. I would suggest the authors to solve all the reviewer concerns, for example regarding readability and clarity of the method and problem, revisiting the claims (or empirical validation to support the claims) and considering to integrate important results requested by the reviewers in the main manuscript.

**Audience:**

Yes

**Audience Explanation:**

Yes, the target task/problem could be of interest for the TMLR audience.

**Claims And Evidence:**

No

**Claims Explanation:**

This work presents CPT, a novel hybrid method integrating In-Context Learning (ICL) with optimization-based methods, with the goal of combining the benefits from both worlds. In particular, the proposed method enhances ICL with a tailored strategy to mitigate overfitting, which is identified as a main limitation of optimization-based approaches within few-shot regimes.

While the reviewers acknowledge the quality of the proposed method and experiments presented, they believe that many concerns raised during the review process have not been addressed, including claims made in the main paper. For example, one of the claims made is that combining ICL with optimization-based methods can leverage the power of ICL to prevent overfitting under extremely low-labeled samples settings. Nevertheless, the proposed method heavily relies on carefully designs to prevent overfitting, from where it cannot be demonstrated whether the claimed benefits come from either resorting to ICL or these carefully designed choices.

Furthermore, several key concerns raised by different reviewers appear to remain unresolved.

**Resubmission Of Major Revision:**

The authors may consider submitting a major revision at a later time.